# Molecular insights into antibody-mediated protection against the prototypic simian immunodeficiency virus

Fangzhu Zhao [1,2,3,13], Zachary T. Berndsen[2,3,4,13], Nuria Pedreño-Lopez [5,13], Alison Burns[1,2,3], Joel D. Allen [6], Shawn Barman[1,2,3], Wen-Hsin Lee [2,3,4], Srirupa Chakraborty[7], Sandrasegaram Gnanakaran[7], Leigh M. Sewall[2,3,4], Gabriel Ozorowski [2,3,4], Oliver Limbo [2,8], Ge Song [1,2,3], Peter Yong [1,2,3], Sean Callaghan [1,2,3], Jessica Coppola[1,2,3], Kim L. Weisgrau[9], Jeffrey D. Lifson[10], Rebecca Nedellec[1,2,3], Thomas B. Voigt [5], Fernanda Laurino[5], Johan Louw[5], Brandon C. Rosen[5,11], Michael Ricciardi [5], Max Crispin [6], Ronald C. Desrosiers[11], Eva G. Rakasz [9], David I. Watkins[5], Raiees Andrabi [1,2,3] ✉, Andrew B. Ward [2,3,4] ✉, Dennis R. Burton [1,2,3,12] ✉ & Devin Sok[1,2,3,8] ✉

SIVmac239 infection of macaques is a favored model of human HIV infection. However, the SIVmac239 envelope (Env) trimer structure, glycan occupancy, and the targets and ability of neutralizing antibodies (nAbs) to protect against SIVmac239 remain unknown. Here, we report the isolation of SIVmac239 nAbs that recognize a glycan hole and the V1/V4 loop. A high-resolution structure of a SIVmac239 Env trimer-nAb complex shows many similarities to HIV and SIVcpz Envs, but with distinct V4 features and an extended V1 loop. Moreover, SIVmac239 Env has a higher glycan shield density than HIV Env that may contribute to poor or delayed nAb responses in SIVmac239-infected macaques. Passive transfer of a nAb protects macaques from repeated intravenous SIVmac239 challenge at serum titers comparable to those described for protection of humans against HIV infection. Our results provide structural insights for vaccine design and shed light on antibody-mediated protection in the SIV model.

Since its first description in 1985, Simian Immunodeficiency Virus (SIV) infection of monkeys has been routinely used as a model for HIV infection of humans[1–4]. The molecular clone, SIVmac239, that infects rhesus macaques has been widely used in vaccination and protection studies[5–10]. The efficacy of several vaccine constructs and vaccine regimens against mucosal challenge by SIVmac239 in macaques has preceded their investigation in HIV vaccine clinical trials. Many of these SIV vaccines have focused on inducing T cell responses alone or T cell responses in combination with non-neutralizing antibody responses. The most successful of the macaque vaccines has been the rhesus cytomegalovirus (RhCMV)-SIV protocol that induces MHC class E-restricted CD8[+] T cells and results in early complete arrest of SIVmac239 replication and subsequent viral clearance in 50–60% of animals[11,12].

In contrast to T cell responses, limited studies have investigated nAb responses to SIVmac239 and their potential protective activities. This paucity likely reflects the observation that nAb responses in natural SIVmac239 infection are relatively weak or delayed, such that isolating monoclonal nAbs is difficult[13,14]. Furthermore, there are no immunogens that have been described as capable of inducing even

autologous nAbs to SIVmac239. In contrast, monoclonal nAbs isolated from HIV infection and SHIV-infected macaques have been critical in establishing the conditions for protection by nAbs against HIV in humans and small animal models and against chimeric simian-human immunodeficiency virus (SHIV) in macaques[15–18].

There is strong and increasing interest in combining T cell and nAb responses in an HIV vaccine[19,20]. This interest arises from the partial success of the RhCMV-SIV strategy described above and the results from the HVTN703/HPTN081 and HVTN704/HPTN085 Antibody Mediated Protection (AMP) study that show relatively high nAb titers are required for passive antibody protection against HIV challenge[21]. The synergy between the two arms of the immune system could lead to more effective and complete protection than achieved with one arm only. Indeed, synergy has been reported for protection against SHIV challenge in macaques; a reduction in the serum nAb titer associated with protection was noted in the presence of a T cell response induced by immunization with a heterologous viral vector regime[22]. As above, however, although improvements have been made to the SHIV model[23–25], the SIV model is the one that has been most explored and is believed the best mimic of the diversity that arises through HIV infection in humans[26]. Thus, to fully explore T cell and nAb synergy in the SIVmac239 model requires nAbs and/or immunogens able to induce nAbs to the virus.

Here, we report the isolation of 12 potent SIVmac239 monoclonal nAbs from three chronically infected rhesus macaques and map their binding specificities on the SIVmac239 Env trimer. We show that passive transfer of one of the nAbs, K11, protects against SIVmac239 repeat intravenous challenge in rhesus macaques with one 50% animal infectious dose ($AID_{50}$). We also present the high-resolution structure of a recombinant SIVmac239 trimer in complex with nAb K11 by electron cryo-microscopy (cryo-EM) and find that it has many features in common with the HIV-1 and SIVcpz Env trimer structures solved to date, along with some distinctive features. To complement the cryo-EM structure, we also determine the identity of all N-linked glycans via mass spectrometry (MS) and use this combined data to carry out computational modeling of fully glycosylated trimers. Our results corroborate the role of the glycan shield in limiting the elicitation of nAbs, highlight the importance of nAbs in protection against SIV infection, provide a structure for the nAb SIV vaccine target and pave the way for studying the combination of T cells and antibodies in the SIVmac239 challenge model.

## Results

### Isolation of neutralizing antibodies from SIVmac239-infected rhesus macaques

We screened infected rhesus macaques for plasma neutralization activity against the SIVmac239 pseudovirus (Supplementary Fig. 1a)[13] and selected three macaques (r11039, r11008, and r11002) with the most potent activity for nAb generation. The animals had been infected for 6–10 months. First, IgG+ memory B cells from peripheral blood mononuclear cells (PBMCs) were antigen-selected using a recombinant SIVmac239 SOSIP.664 trimer[27] (Fig. 1a and Supplementary Fig. 1b). An average of 1.62% of CD20+IgM−IgG+ memory B cells bound to the SIVmac239 trimer antigen. Antigen-specific B cells were single-cell sorted and cultured in 384-well plates[28,29]. A total of 8470 single cells were sorted and expanded in vitro, from which 1546 wells secreted detectable IgG in supernatants. Around 52% of the IgG+ wells showed measurable binding to SIVmac239 SOSIP protein by enzyme-linked immunosorbent assay (ELISA) screening, but only 20 wells were positive for SIVmac239 pseudovirus micro-neutralization. The frequency of autologous nAb hits identified by supernatant screening for SIVmac239 was lower than SHIV$_{BG505}$ infected macaques (0.26% and 1.18% using 80% neutralization cut-off, based on SIVmac239 SOSIP vs BG505 SOSIP sorting, respectively) (data adapted from ref. 29). Hence, the

majority of SIV Env-specific memory B cells target non-neutralizing epitopes, likely more so than in SHIV or HIV infection[30].

We cloned and expressed 12 monoclonal antibodies (mAbs) and annotated them to the rhesus macaque germline database[31], where we identified three clonally related lineages, numbered 1–3 (Fig. 1b). The SIV mAbs have an average of 10.8% somatic hypermutation (SHM) in the heavy chain (HC) and 6.4% in the light chain (LC) at the nucleotide level (Supplementary Fig. 1c). The third complementarity determining region of the heavy chain (CDRH3) in lineages 1 and 2 is relatively long, at 25 and 18 amino acids, respectively, whereas lineage 3 has a more average CDRH3 length of 13 amino acids (Fig. 1b)[32,33].

All 12 mAbs neutralized the tier 3 SIVmac239 pseudovirus in the TZM-bl assay (Fig. 1c), with the most potent antibody FZ019.2 neutralizing at an $IC_{50}$ value of 0.005 µg/mL (Fig. 1d). However, lineage 3 antibodies FZ019.2 and J9 showed incomplete neutralization (90% and 72%, respectively) at the highest antibody concentration (25 µg/mL) (Fig. 1 c, d). In contrast, antibodies in lineages 1 and 2 fully neutralized SIVmac239 at a median $IC_{50}$ value of 0.40 µg/mL (Fig. 1 c, d). The lineage 1 K11 antibody neutralized SIVmac239 at an $IC_{50}$ of 0.12 µg/mL, which is 5-fold more potent than ITS90.03, the only reported SIVmac239 nAb lineage so far[34] (Fig. 1d).

### Rhesus nAbs target a glycan hole epitope and the V4 loop on the SIVmac239 Env

To map the binding specificity of isolated mAbs, we performed ELISA and included the non-neutralizing antibody (non-nAb) 5L7 for comparison[9]. All mAbs from lineages 1 and 2 bound to SIVmac239 SOSIP.664, SIVmac239 gp120, SIVmac239 gp140 foldon trimer (FT)[34], and gp140 FT dV1V2V3 core proteins, whereas FZ019.2 and J9 showed decreased binding activity against gp120 dV1V2 and gp140 FT dV1V2V3 proteins (Supplementary Table 1).

Competition ELISA revealed that antibodies in lineages 1 and 2 target a similar epitope as ITS90.03 (Supplementary Fig. 2a). ITS90.03 has previously been shown to recognize a glycan hole on SIVmac239 gp120 Env centered on K254 (numbered by SIVmac239, 238 numbered by HIV HxB2)[34]. Thus, it appears that the glycan hole is a notable neutralizing epitope among SIVmac239-infected rhesus macaques. The lineage 3 nAbs, FZ019.2 and J9, did not compete with ITS90.03, nor did they compete with CD4-IgG2. Of note, the SIVmac239 SOSIP trimer that we used contains a K180S substitution in the V2 loop to increase the binding affinity of the V2 apex-specific HIV-1 broadly neutralizing antibody (bnAb) PGT145, which enables its use in competition experiments[27,35]. The lineage 3 nAbs also did not compete with PGT145 for Env trimer binding, suggesting a distinct epitope that is not V2 apex or CD4 binding site (CD4bs) (Supplementary Fig. 2a). Evaluation of lineage 3 antibodies against glycosidase-treated SIVmac239 pseudovirus in neutralization assays indicated a dependence of these antibodies on hybrid-type or complex-type glycans to some degree (Supplementary Fig. 2b).

We used single-particle negative-stain electron microscopy (nsEM) to further characterize antibody epitope specificities by complexing with SIVmac239 SOSIP.664 Env trimer. The 3D reconstructions revealed three K11 antibody molecules bound to the trimer (Fig. 2a). Consistent with the competition ELISA data, K11 recognized a similar glycan hole epitope as ITS90.03[34]. ITS90.03 failed to neutralize the SIVmac239 K254N mutant, which creates the N-link glycan sequon (Asn-X-Ser/Thr, where X is any amino acid except proline) at that position. Similarly, K11 showed dramatically decreased neutralization activity against this virus mutant (Fig. 2b). By contrast, neutralization of FZ019.2 was not affected by mutations at the 254 position (Fig. 2b), and nsEM revealed that FZ019.2 binds near the V4 loop on SIVmac239 Env (Fig. 2c, d and Supplementary Fig. 3a, b). Alanine-scanning mutagenesis revealed that several V4 residues, including N415, T416, N418, Q419, and P421 are important for FZ019.2 neutralization activity (Supplementary Fig. 2c).

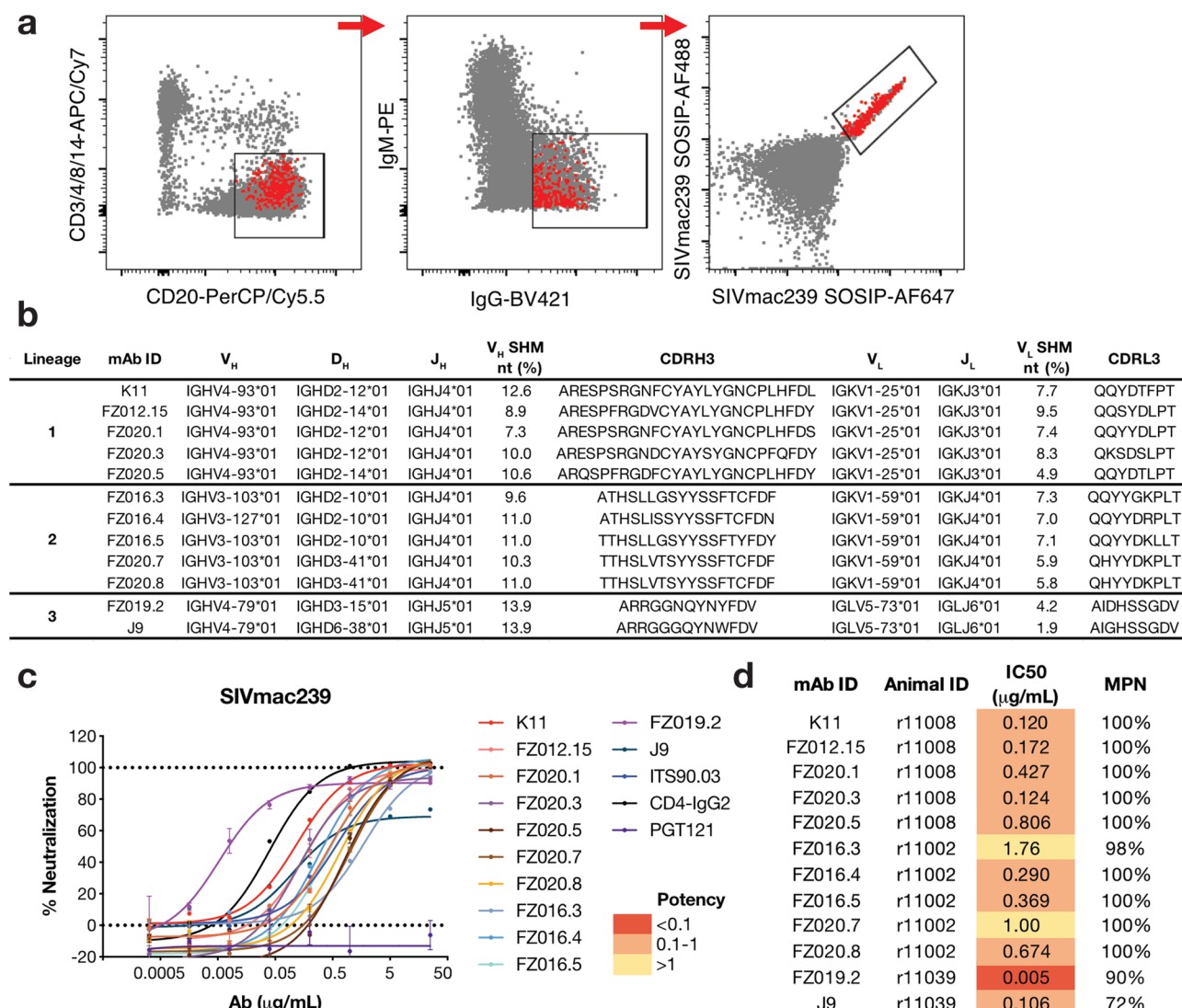

**Fig. 1 | Isolation of SIVmac239 neutralizing rhesus mAbs. a** FACS sorting layout from r10039 rhesus PBMCs, in which population is gated on lymphocytes/singlets. CD3⁻CD4⁻CD8⁻CD14⁻CD20⁺IgM⁻IgG⁺ SIVmac239 SOSIP.664²⁺ memory B cells were single-cell sorted. Red dots represent sorted cells. **b** Immunogenetics of isolated rhesus mAbs. Rhesus mAbs were annotated with a rhesus germline database[31].

**c**, **d** SIVmac239 pseudovirus neutralization curves. Data are presented as mean values ± SD and error bars are from two technical replicates. Data are representative for at least three independent experiments. **c** and summary table **d** of neutralization IC₅₀ and maximum percentage of neutralization (MPN) for denoted mAbs. IC₅₀ potency was colored according to the key.

We further tested the neutralization breadth of mAbs against a panel of SIV pseudoviruses[34,36]. As expected, the neutralization breadth of glycan hole-targeting antibodies depended on the residue at position 254 on gp120: most viruses harboring the glycan hole were sensitive to neutralization, while other viruses with an N-linked glycan sequon at position 254 were more neutralization resistant (Fig. 2e). In contrast, FZ019.2 and J9 nAbs only neutralized autologous SIVmac239 and SIVmac239.cs.23 isolates (Fig. 2e), which is as expected since the V4 loop is poorly conserved between isolates.

**Non-neutralizing SIVmac239 epitopes are immunodominant**
To assess epitope specificities for non-nAbs to SIVmac239, we isolated seven mAbs that bound SIVmac239 Env proteins but did not neutralize the autologous virus (Supplementary Fig. 2d, e). We similarly performed ELISA epitope mapping and competition assays. These data reveal that the majority of the non-nAbs recognize a similar specificity to 5L7 (Supplementary Fig. 2d). These antibodies target an epitope that is not dependent on the V1, V2, and V3 loops on gp120 (Supplementary

Fig. 2f). nsEM indicates that the prototype FZ012.7 antibody binds near the V4 and V5 loops (Supplementary Fig. 3e). Antibodies FZ012.14 and FZ012.16 appear to recognize the V3 loop on SIVmac239 Env as they bind SIVmac239 gp120 but fail to bind SIVmac239 dV3 protein (Supplementary Fig. 2f).

To further elaborate on the antigenic surface of SIVmac293 Env, we mapped the epitopes targeted in polyclonal serum responses among 9 infected rhesus macaques. As shown in Supplementary Fig. 1a, 4 macaques developed nAb responses against SIVmac239 (r11008, r11002, r11039, and r11004) and five displayed undetectable serum nAb titers (Rh33519, Rh31186, Rh34620, Rh32388, and Rh34118). All animals showed serum binding titers to SIVmac239 SOSIP.664. We performed serum competition assays with SIVmac239 SOSIP.664 as antigen and observed that all SIVmac239-infected macaque sera strongly competed for binding with antibodies that recognize the 5L7/FZ012.7 epitope and a few animals competed for binding to the V3 loop (Fig. 3a). None of the sera showed strong competition with PGT145, CD4-IgG2, FZ019.2, or K11 for binding to SIVmac239

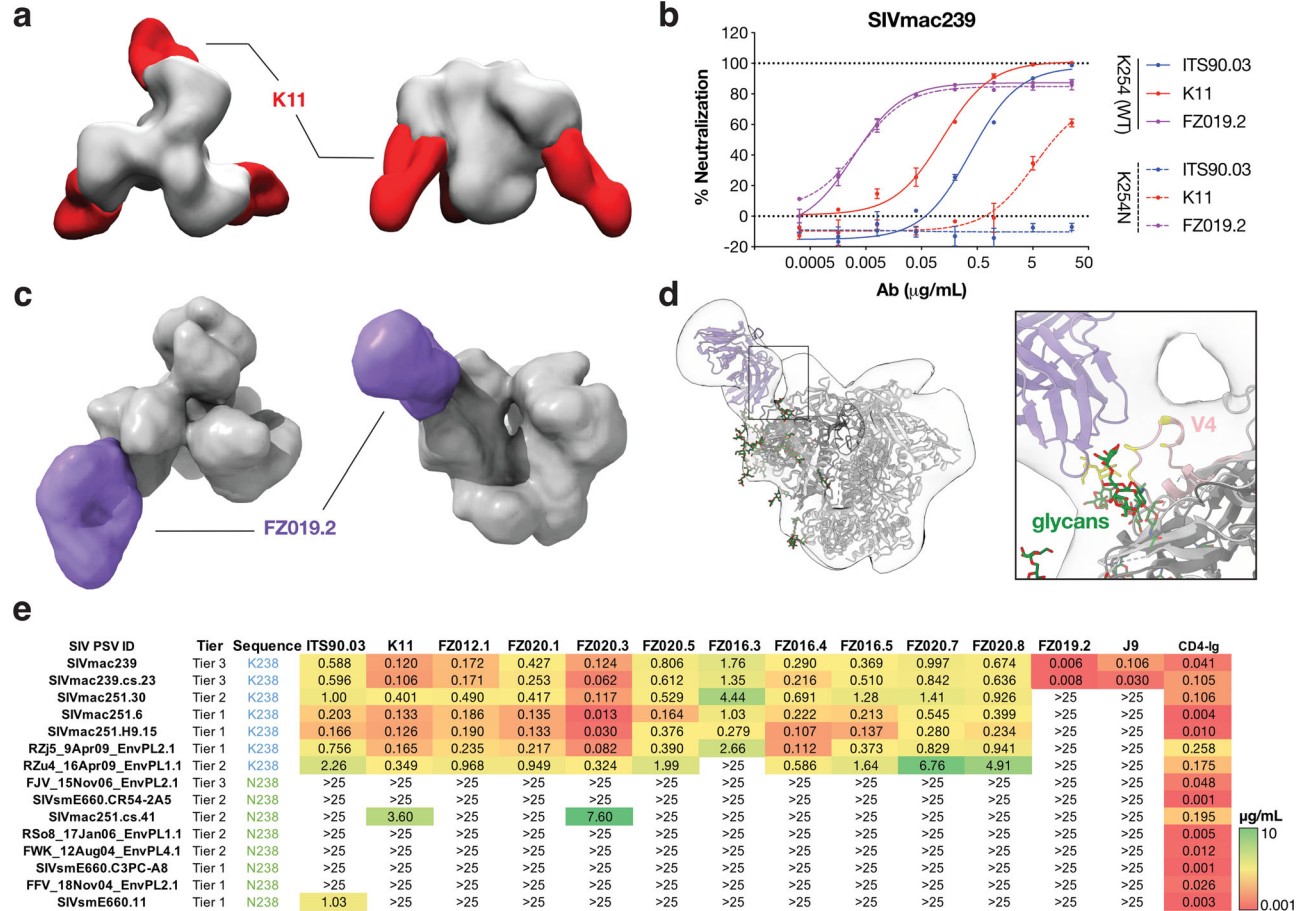

**Fig. 2 | SIVmac239 nAbs recognize two specificities: a glycan hole and the V4 loop. a** Negative-stain electron microscopy (nsEM) of K11 (red) bound to SIV-mac239 SOSIP.664 (gray). **b** Representative neutralization curves of K11 (red), ITS90.03 (blue), and FZ019.2 (purple) against wild-type SIVmac239 virus (solid line) and K254N mutant with N-linked glycan filled in (dashed lines). Data are presented as mean values ± SD and error bars are from two technical replicates. Data are representative for at least two independent experiments. **c** nsEM reconstruction of

FZ019.2 Fab (purple) bound to SIVmac239 SOSIP.664. **d** Model of the complex using BG505 SOSIP.664 (PDB:6X9R) with the SIVmac239 delta V1/V2 gp120 monomer crystal structure (PDB:6TYB) aligned to one protomer showing that FZ019.2 targets the V4 loop (pink). Residues within 5 Å of the Fab are colored yellow. **e** Neutralization breath and potency (μg/mL) of isolated mAbs against an SIV pseudovirus panel. CD4-IgG2 served as positive control. IC$_{50}$ potency was colored according to the key.

SOSIP.664, suggesting that the neutralizing epitopes are relatively poorly targeted in natural infection and consistent with relatively low serum neutralizing titers.

Additionally, all the SIVmac239-infected macaque sera bound overlapping SIVmac239 Env 15-mer peptides on V1V2, V3, and gp41 (Fig. 3b and Supplementary Table 2), whereas non-infected serum and SIVmac239 nAbs did not. We further analyzed the polyclonal antibody binding specificities in SIVmac239-infected macaque sera with electron microscopy polyclonal epitope mapping (EMPEM)[37]. 2D classification revealed a high proportion of non-trimeric Env fragments that were extensively decorated with polyclonal Fabs (Fig. 3c and Supplementary Fig. 3f). These degraded trimers could result from the inherent instability of SIVmac239 SOSIP.664 and/or antibody-induced trimer disassembly, a phenomenon recently observed in HIV Env wherein certain gp41 antibodies can induce trimer disassembly[38]. Here, we observe that non-trimer particles are more decorated with Fabs than the intact trimers, consistent with some of the polyclonal Fabs being specific to these nonfunctional forms of Env. This is furthermore consistent with observations that polyclonal sera interact with non-functional forms of Env[39], the large discrepancy between ELISA and neutralization[40], and the similarity in the peptide fragment binding data between the neutralizing and non-neutralizing macaque sera (Fig. 3b). Of the small minority of stable Fab-bound trimer classes we could reliably reconstruct in 3D, we found that most non-neutralizing

sera contained gp41 specificities similar to the HIV-1 non-nAb classes known to cause trimer disassembly[38]. One dataset yielded a class with potential V1/V4 specificity similar to FZ019.2, however, it was a rare class with relatively few particles and likely represents a minor species within the larger polyclonal pool. Of the neutralizing sera, two of three yielded classes with clear glycan hole specificity and one with FZ019.2-like and gp41 specificity. No V1/V2-binding response was observed in EMPEM, possibly due either to a low abundance of antibodies to this epitope region or disassembly of the SIVmac239 trimer by V1/V2-binding antibodies.

## Cryo-EM structure of SIVmac239 Env trimer in complex with the nAb K11

To date, there is only structural information available for the SIVmac239 delta V1/V2 gp120 monomer[34] and no structure for a native-like SIVmac239 trimer. Our previous attempts to solve the trimer structure via cryo-EM failed due to severe orientation bias and trimer instability[27]. However, by complexing the trimer with K11 IgG, we were able to overcome these issues and obtain a 3.4 Å-resolution reconstruction of the complex (Fig. 4, Supplementary Fig. 4a–i, and Supplementary Table 3). Overall, the topology of the trimer closely resembles both the chimpanzee SIVcpz (MT145K) and HIV-1 Env (BG505), with the V2 and V3 loops forming the trimerization interface at the membrane-distal apex (Fig. 4A, B). Alignment of the previously

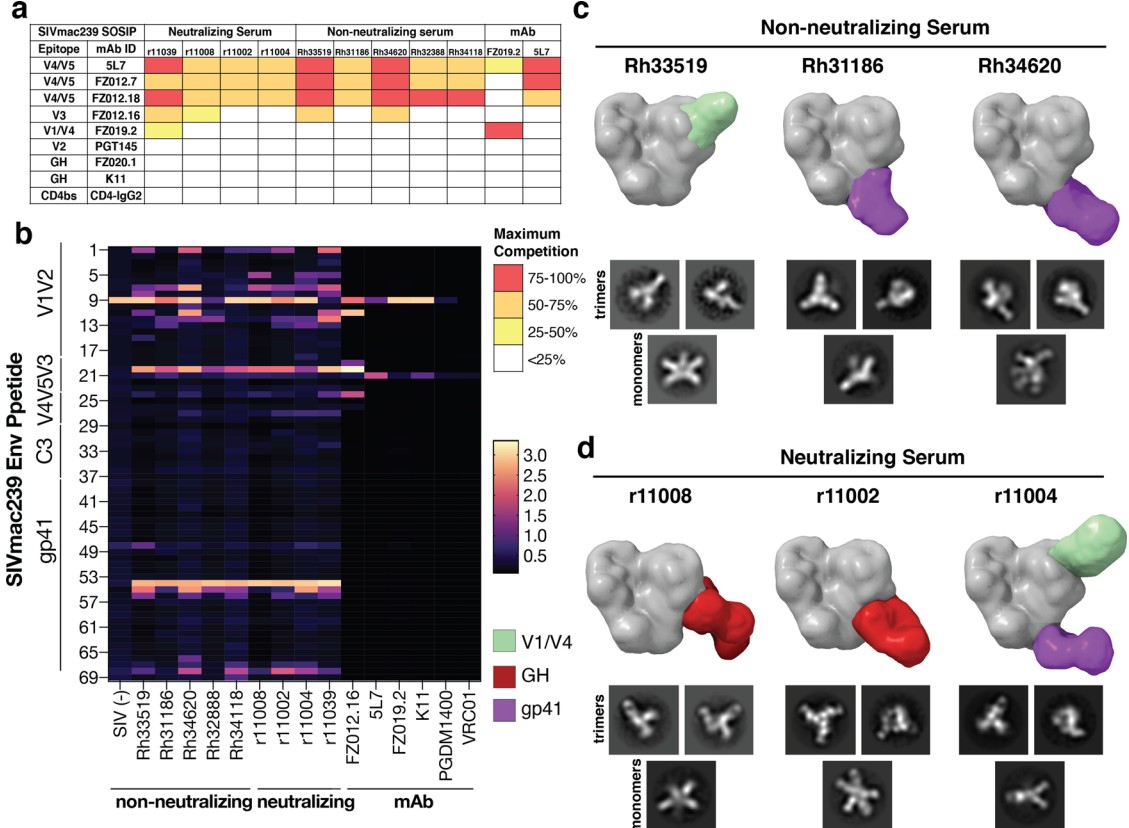

**Fig. 3 | Mapping of polyclonal antibody epitope specificities. a** ELISA competition between a panel of SIVmac239 Env-binding mAbs and SIVmac239-infected macaque sera for binding to SIVmac239 SOSIP.664. Maximum percentage of competition is colored according to the key. **b** ELISA screening of infected rhesus sera to SIVmac239 Env 15-mer peptides. Peptide ID numbers were listed on the left and sequences were shown in Supplementary Table 2. SIVmac239 neutralizing mAbs (K11, FZ019.2), non-neutralizing mAbs (FZ012.16, 5L7), and HIV bnAbs (PGDM1400, VRC01) served as controls. Absorbance intensities at OD405 nm were colored according to the key. **c, d** EMPEM mapping of SIVmac239-infected **c** non-neutralizing macaque sera Rh33519, Rh31186, Rh34620 and **d** neutralizing macaque sera r11008, r11002, r11004 for binding to SIVmac239 SOSIP.664. Data shown are representative 2D class averages of trimeric and monomeric particles bound to polyclonal Fabs along with segmentations of 3D reconstructions showing the location of bound Fabs relative to a molecular surface model of the SIVmac239 structure. Epitope specificities are colored according to the key.

solved crystal structure with our cryo-EM structure reveals a close fit in the gp120 core (root mean squared deviation (RMSD) = 0.67 Å), but as expected, significant deviation around the gp41 and trimerization interfaces (full model RMSD = 3.65 Å; Supplementary Fig. 5a). The two features of the SIVmac Env structure that are most distinct from HIV and SIVcpz are the V4 loop and extended V1 loop which both fold back on top of the gp120 core (Fig. 4B). These conformations are possible because of differences in glycosylation between SIVmac239 and both HIV and SIVcpz (Fig. 4C). If we map the distinct SIVmac239 features on HIV and SIVcpz Env, the extended V1 loop would clash with the conserved HIV high-mannose patch glycans at positions N332 and N295 and the SIVcpz V4 glycan at N412, while the V4 loop would clash with the gp120 outer-domain glycans at N363, N386, and N392 on both HIV and SIVcpz (Fig. 4C), the latter two of which are highly conserved among HIV strains. To shield parts of the extended V1 loop, SIVmac239 has two glycans at positions N146 and N156 (Fig. 4C, D). The majority of the extended V1 loop is well resolved in our cryo-EM map except for the stretch between residues 127 and 139, which appears to extend upward away from the gp120 core (Fig. 4D, Supplementary Fig. 4k). Overall, HIV BG505 SOSIP.664 is more similar to SIVcpz MT145K than SIVmac239, with RMSDs of 8.11 and 9.79 Å, respectively (Supplementary Fig. 5b). Other regions of interest that show structural differences from SIVcpz and HIV are in the V5 loop, regions within gp41 HR1 and HR2, and the fusion peptide (FP; Supplementary Fig. 5c). The FP is buried against the gp120 core, unlike in HIV where it protrudes outwards, and SIVcpz where it is buried even further behind the fusion

peptide proximal region helix (Supplementary Fig. 5c). SIVmac239 also features an additional disulfide bond between residues 194 and 206 in the V2 loop (Supplementary Fig. 5d).

Our cryo-EM structure confirms that K11 targets the same glycan hole as ITS90.03 (Fig. 4E), which is located in a position equivalent to the well-characterized 241/289 glycan hole on BG505 SOSIP.664. In addition to the absence of a glycan sequon at position 254, which has a PNGS in ~80% of other SIVmac sequences[34], another PNGS at N247 also sits within both the K11 and ITS90.03 epitopes and is almost completely unoccupied (Fig. 4A, E). That both antibodies depend on this site being unoccupied for virus neutralization suggests the sub-occupancy is not unique to the soluble SOSIP trimer and must also occur on the native spike.

The K11 epitope-paratope interactions are largely dominated by sidechain-backbone hydrogen bonds between the CDRH3 loop and gp120 residues, as well as hydrogen bonds with the unique SIVmac239 glycan at N37, which interacts with both the HC and LC (Fig. 4F). The K11 CDRH3 loop is 25 residues in length and features an interloop disulfide bond between residues $C100_C$ and $C100_K$ (Fig. 4F). The glycans at positions N284, N295, and N371 also surround the epitope but are not engaged directly. Although the ITS90.03 and K11 epitopes overlap extensively (Fig. 4E), the orientation of the two antibodies is orthogonal to one another, with K11 and ITS90.03 being oriented parallel and perpendicular to the viral membrane surface, respectively (Fig. 4F), which expands the binding footprint of ITS90.03 into the gp120-gp41 interface region. The containment of the epitope of K11

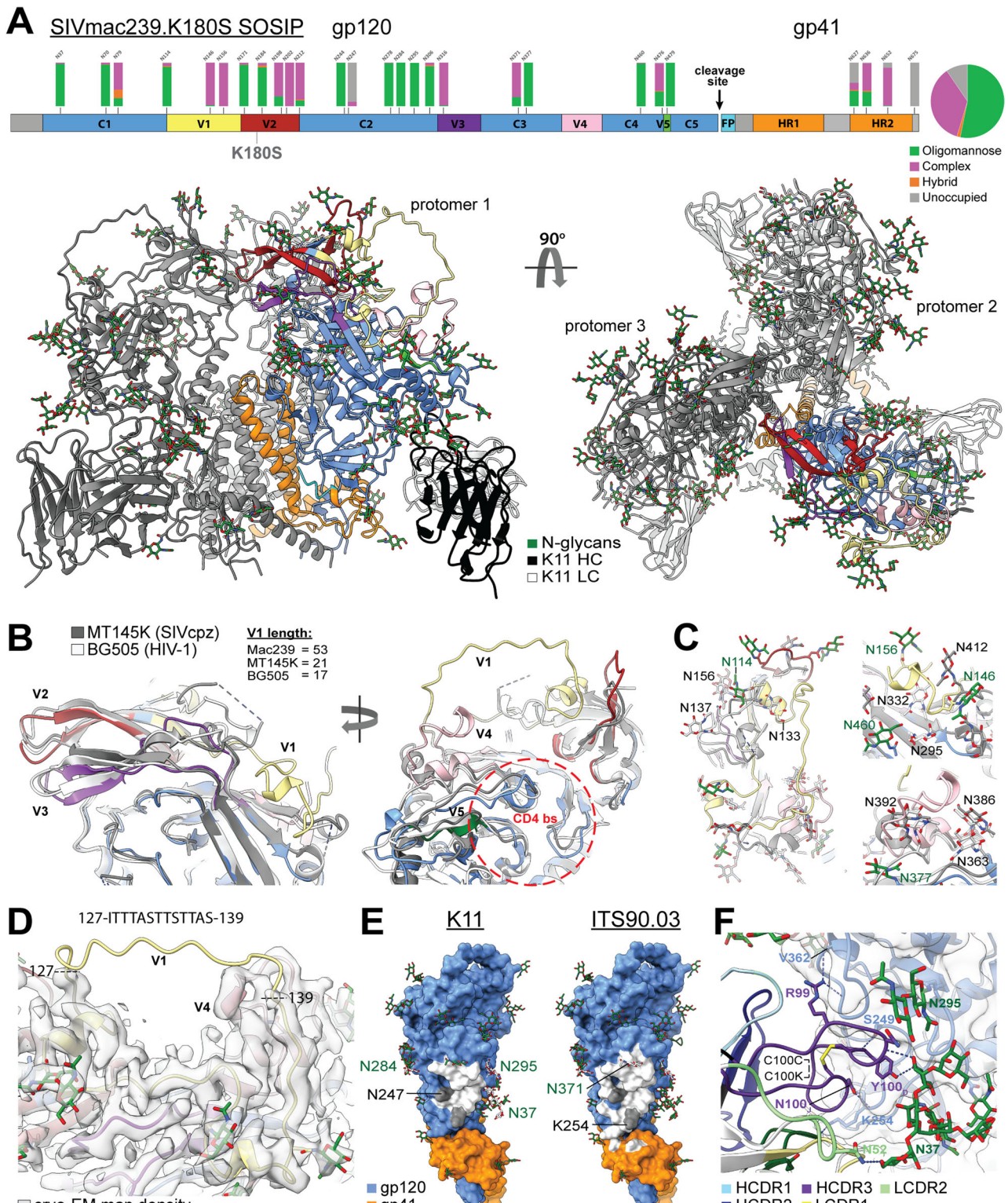

**Fig. 4 | Cryo-EM structure of SIVmac239.K180S SOSIP trimer in complex with the nAb K11. A** Domain organization, site-specific N-linked glycan distribution, and cryo-EM structure of SIVmac239.K180S SOSIP.664 in complex with K11. **B** Close-up of the V1-V5 loops with the structures of SIVcpz MT145K (PDB:6OHY) and HIV-1 BG505 SOSIP (PDB:6X9R) Env trimers overlayed for comparison–glycans not shown. **C** Close-up views of the novel SIVmac239 V1 and V4 loop conformations showing the positions of N-linked glycans (Asn-NAG only) for all three structures to emphasize the clashes that would occur with HIV and SIVcpz glycans. **D** Cryo-EM map density for the novel V1 and V4 loop conformations with the sequence of the disordered V1 fragment indicated above. **E** Epitopes of both K11 and ITS90.03 (PDB:6TYB) mapped to the SIVmac239 trimer structure (white indicates residues within 5 Å of Fab residues). **F** Close-up view of the K11 epitope-paratope showing hydrogen bonds with gp120 residues and N-linked glycans. Also indicated is the intra-CDRH3 disulfide bond.

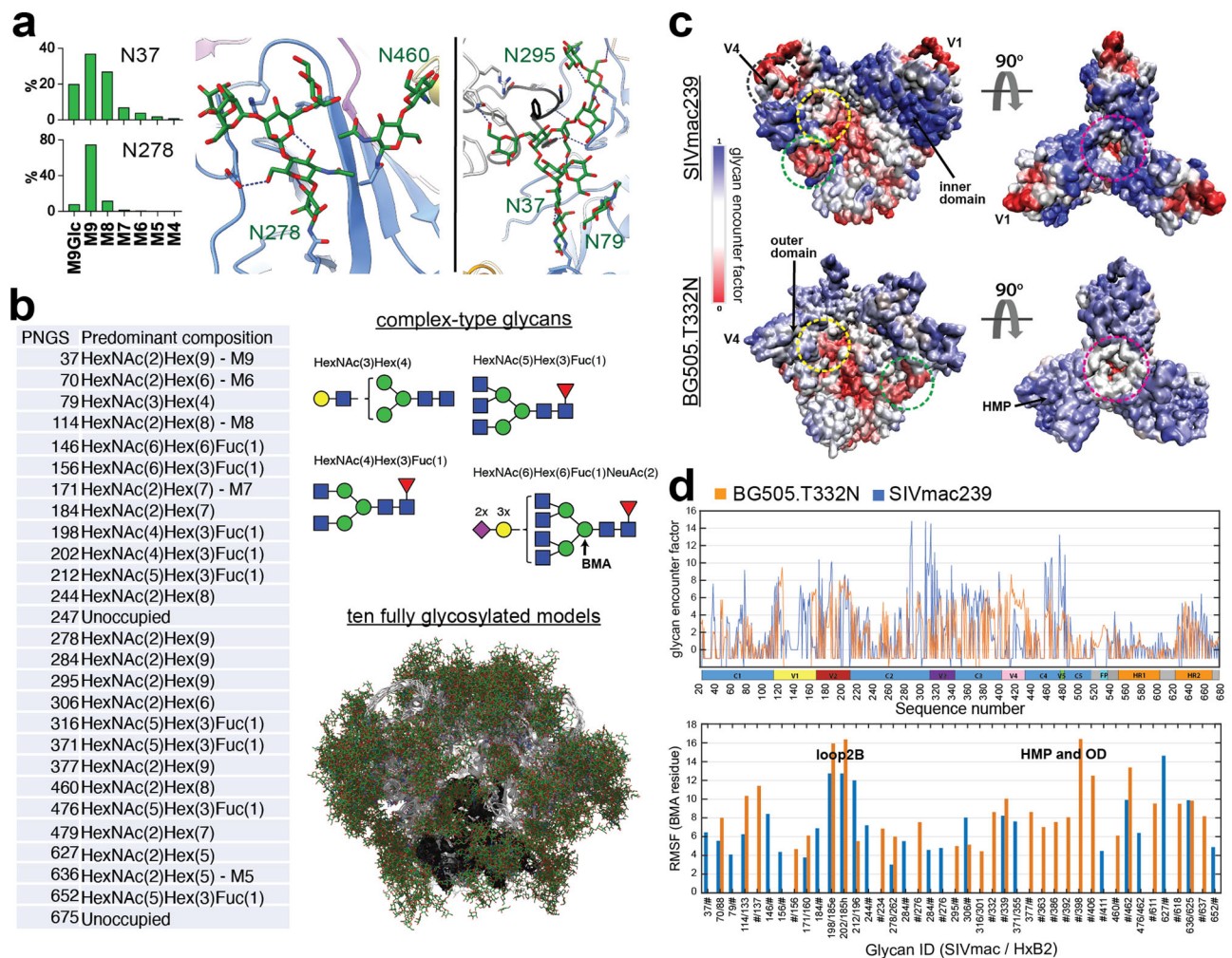

**Fig. 5 | Site-specific MS and computational modeling of the SIVmac239 Env trimer glycan shield. a** Predominant glycoforms at each site as determined by MS used for computational modeling along with cartoon representations of each type of complex glycan and a visualization of 10 (1000 total) different fully glycosylated models with glycans in green and gp120 and gp41 in white and black respectively. **b** Glycoform distribution for the two highly under-processed glycan sites at N37 and N278 showing the detection of glucosylated Man9 at both sites along with views of each glycan structure showing stabilizing hydrogen bonds with neighboring protein and glycan residues. **c** Glycan encounter factor calculated for every surface-exposed residue using 1000 fully glycosylated models of both SIVmac239

and BG505 SOSIP.664 and projected as a color map onto the molecular surface representation. Blue indicates the most shielded surfaces and red the most exposed. Demarcated with dashed circles are some prominent regions including common neutralizing epitopes such as the CD4 binding site (yellow), glycan hole (green), V2 apex (magenta), and the V1/V4 epitope identified in this study (black). Other regions of interest such as the gp120 inner and outer domains (OD), V1 and V4 loops, and high-mannose patch (HMP) are labeled. **d** Per-residue glycan encounter factor and per-glycan root mean squared fluctuation (RMSF) calculated from 1000 fully glycosylated trimer models.

---

within gp120, as opposed to the epitope of ITS90.03, which also includes gp41, might result in more potent antibody neutralization, as the gp41 and the gp120-gp41 interface are more flexible than the gp120 core.

## The glycans on SIVmac239 Env are more densely packed and shield more protein surface area than on HIV Env

Autologous nAb responses to SIVmac239 induced by infection are relatively weak compared to typical responses observed in HIV infection[13,14,41–43]. Since neutralization has been correlated with binding to native Env, we considered whether structural features of the SIVmac239 Env might provide clues as to relatively low immunogenicity. Indeed, we found that the glycans on SIVmac239 Env are more closely packed and shield more protein surface than on HIV Env.

We determined the identity of all N-linked glycans on Env via site-specific mass spectrometry (Supplementary Fig. 6a, b). The proportion of potential N-linked glycosylation sites (PNGS) containing high-mannose, complex-type, or no glycan at all are displayed above the

sequence at each of the 27 PNGS (Figs. 4a, 5a). Similar to HIV and SIVcpz Env trimers[44,45], many of the sites contain oligomannose-type glycans (13 in total; greater than 80% of the total glycoforms observed) which indicate that, for the recombinant SIVmac239 Env trimer, these glycans are under steric constraints that prevent complete enzymatic processing. This can arise because of crowding from neighboring glycans or constraints from the surrounding protein structure[46]. Six sites present a mixture of glycan processing states, both oligomannose and fully processed complex-type glycans, including two of the glycan sites located on gp41. Fully processed complex-type glycans are present on 8 sites on both gp120 and gp41, and these correspond to sites that are easily accessible to processing enzymes. This distribution of glycans is reminiscent of HIV and SIVcpz Env with two notable exceptions: the loss of gp120 outer-domain and high-mannose patch glycans due to the V1 and V4 loop conformations and the presence of a small fraction of glucosylated Man9GlcNAc2 (Man9) glycans.

The presence of glucosylated Man9 has important implications. Prior to the initial mannosidase trimming in the ER and Golgi, the Man9

glycan contains 1–3 glucose residues that act as a signal to the calnexin/calreticulin folding machinery that the protein has not folded correctly. Normally the enzyme that removes these glucose residues can readily access even those glycans that are restricted from further processing[47] but glucosylated structures can still be secreted at sterically protected sites[48]. Their presence could indicate that the glycan shield on SIVmac239 is even more densely packed than on either SIVcpz or HIV Env. There are two sites with detectable levels of monoglucosylated Man9, namely N37 and N278[48] (equivalent to the N262 glycan on HIV when aligned with HxB2), with the former having a higher percentage of monoglucosylated Man9 (Fig. 5b). Both glycans engage in multiple stabilizing hydrogen bonds with surrounding protein residues (Fig. 5b), which could restrict access of processing enzymes, and suggests they are likely structurally important. Both sites are surrounded by several other glycans and N37 is even further stabilized by inter-glycan hydrogen bonds (Fig. 5b). However, a caveat should be noted; because K11 also stabilizes the N37 glycan, we cannot say if these interactions would be retained in the absence of the antibody.

To quantify the extent of glycan shielding across the trimer surface and the steric restriction of individual glycans, we used a slightly modified version of our previously published high-throughput atomistic modeling (HTAM) pipeline to generate a large ensemble of fully glycosylated SIVmac239 trimer structures[46,49] (See Methods; Supplementary Fig. 6c, d). For comparison, we also performed the same analysis on an equally sized ensemble of fully glycosylated BG505 SOSIP.664 models. By calculating the per-residue glycan encounter factor (GEF), a measure of glycan shielding, we found that overall, SIVmac239 is more strongly shielded than BG505 despite having one less glycan per protomer (Fig. 5c). The gp120 inner domain of SIVmac239 SOSIP.664 is the most strongly shielded while the glycan hole, V1, and V4 loops are the most exposed (excluding the trimer base; Fig. 5c, d). As mentioned previously, the neutralizing antibody FZ019.2 epitope mapped to this region and docking of our cryo-EM structure into the nsEM map indicates that its epitope likely contains segments of V1 as well V4 (Fig. 2c, d and Supplementary Fig. 3c, d). Consistent with the higher level of shielding, the SIVmac239 glycans are more ordered overall (lower root mean squared fluctuation) (Fig. 5d), which is indicative of more glycan-glycan interactions/crowding and/or glycan-protein interactions[46]. The glycan at position N278 (262 in HxB2 numbering) has the lowest RMSF, which is consistent with its high degree of ordering in the cryo-EM map and steric restriction implied by the presence of glucosylated Man9 at this site (Fig. 5b). The glycan at N37, which contains an even higher percentage of glucosylated Man9, has a relatively average RMSF, however, suggesting the restricted access of glycan processing enzymes is due in part to geometric constraints from its occluded position and stabilizing interactions with surrounding residues in addition to crowding from neighboring glycans.

## K11 protects rhesus macaques from repeated SIVmac239 challenge

There are very few studies investigating the ability of antibodies directly to protect against SIVmac239 infection[6,9,50]. Although not strictly an antibody, we showed that CD4-IgG2, a potent neutralizer of SIVmac239, protected against virus challenge at surprisingly low neutralizing titers[6] compared to observations made for nAbs in the SHIV model or indeed in the AMP study[15,21]. One study did show protection in a single monkey against SIVmac239 challenge with the non-nAb 5L7 at relatively high concentration[9]. Accordingly, we investigated the protective efficacies of nAb K11 and non-nAb 5L7 against SIVmac239 in macaques.

K11 was first shown to neutralize the replication-competent SIVmac239 challenge stock at an IC$_{50}$ of 2.2 μg/mL, a 9-fold lower potency than against the pseudovirus (Supplementary Fig. 7a). The challenge study consisted of three groups of rhesus macaques with six animals per group: Group 1: 5L7-LS group (LS mutation in the antibody Fc region to increase half-life[51]), Group 2: K11 group, and Group 3: untreated control group (Supplementary Table 4). K11 or 5L7-LS mAbs were infused by the intravenous route at a dose of 60 mg/kg one day prior to the challenge. All animals were subsequently challenged by the intravenous route with weekly repeat intravenous inoculation with 1 AID$_{50}$ dose of SIVmac239[9] (Fig. 6a). After three repeated weekly challenges, uninfected animals received a second 60 mg/kg dose of antibody, followed by repeated weekly virus challenges.

Following SIVmac239 challenge, 6/6 of control animals and 4/6 of 5L7-LS treated animals became infected after the first challenge (Fig. 6b, c). The remaining two animals r18011 and r18044, in the 5L7-LS group became infected 3 days after the second challenge. In contrast, all animals administered K11 showed protection after two challenges. Two K11-treated animals, r17041 and r18014, with the lowest serum concentrations (Supplementary Fig. 7b), became infected 7 days after the third challenge. Following the second infusion of K11 antibody, animals r18022 and rh2961 became infected after the seventh challenge, and the remaining animal r17034 became infected after the eighth challenge (Fig. 6b, c). Notably, animal r17075 in the K11 group was mistakenly detected as having detectable viral load after the fourth challenge and was therefore not subjected to additional challenges. However, its viral load was undetectable since then, and the viral load repeat for the infection timepoint was negative. Therefore, this animal was excluded from the survival analysis shown in Fig. 6b. None of the 5L7-LS-treated or K11-infected animals showed reduced peak viral load compared with the control group ($P = 0.219$ and $P = 0.160$, respectively, Mann–Whitney test) (Fig. 6d). Of note, K11 significantly suppressed viral load during chronic phase infection ($P = 0.004$) compared with the control group. This suppression was not observed for animals receiving the non-nAb 5L7-LS (Fig. 6e).

Among the infected animals in the K11 group, the geometric mean of plasma SIVmac239 nAb ID$_{50}$ titers was 1:609 at 7 days prior to infection (Fig. 6f and Supplementary Fig. 7c), which is approximately 2-fold higher than the equivalent titer reported from SHIV challenge studies[15,52]. We did not perform a Bayesian logistical regression of serum nAb titer and infection probability as described in Pauthner et al. due to the small number of animals in this study.

To investigate the contribution of the effector function of the antibody, we measured the antibody-dependent cellular cytotoxicity (ADCC) activity, where we observed better killing activity by SIVmac239-specific mAbs than CD4-IgG2 (Fig. 6g). However, the difference in ADCC activity between K11 and 5L7 was not significant.

Taken together, our findings indicate that, in the case of humoral immunity, antibody neutralization activity is necessary and sufficient for protection against SIVmac239 virus challenge.

## Discussion

The phase IIb HVTN703/HPTN081 and HVTN704/HPTN085 trials (also known as the Antibody Mediated Protection study or AMP study) showed that a bnAb can protect humans against HIV exposure, albeit at relatively high serum antibody neutralizing titers[21]. Many studies show that bnAbs can protect non-human primates (NHPs) against SHIV challenge, again at similar relatively high serum antibody titers[15]. To date, there have not been comprehensive studies on the contribution of nAbs to SIVmac239 challenge because nAbs against this virus were not available until recently[34]. For cellular immunity to HIV, T cell responses have clinically failed to protect humans against HIV exposure in a variety of formats[53–56]. Notably, the negative results of the STEP trial were predicted by SIV but not SHIV studies. For the SIV model, CMV vectors that induce MHC-E-restricted T cell responses have demonstrated substantial control of viral replication after SIVmac239 challenge[11,12,57]. Importantly, there is considerable motivation

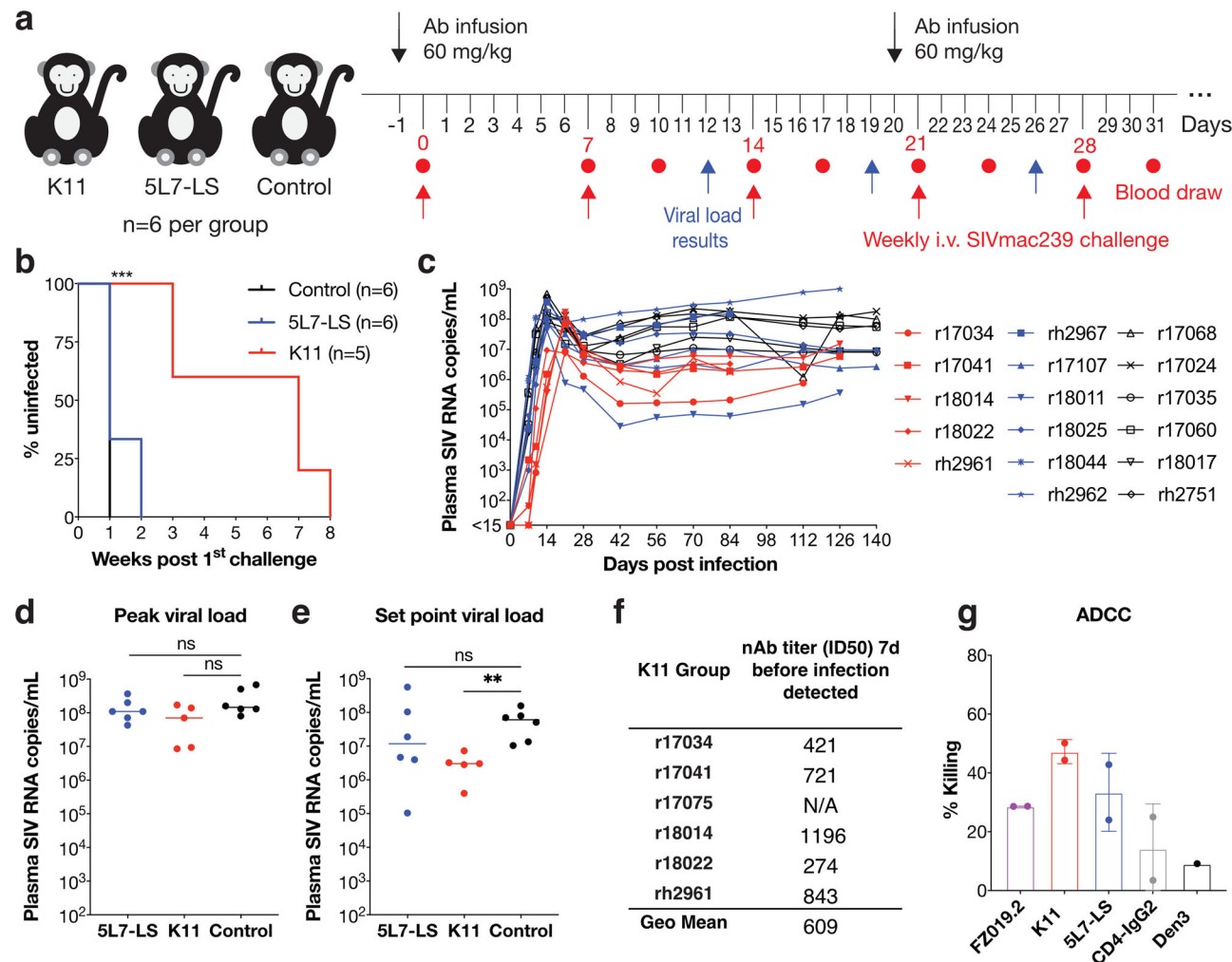

**Fig. 6 | Repeated SIVmac239 intravenous (IV) challenge of 5L7-LS-infused, K11-infused, and control animals. a** SIVmac239 passive Ab protection experimental design. Rhesus macaques were allocated to three groups; Group 1 and Group 2 were infused with 60 mg/kg of 5L7-LS and K11 antibody, respectively, 1 day prior to challenge. Group 3 animals were untreated controls. Animals were challenged weekly by the IV route with a repeat 1 $AID_{50}$ dose of SIVmac239. Blood was drawn and plasma viral load was measured at day 7 and day 10 following each challenge. Animals with detectable viral loads at either of those time points were no longer challenged. After three repeated challenges, a second antibody infusion was given to all Group 1 and Group 2 animals. Uninfected animals were subsequently repeatedly challenged intravenously with 1 $AID_{50}$ dose of SIVmac239. **b** Kaplan–Meier curves comparing three test groups colored according to the key. Protection was measured by log-rank (Mantel-Cox) test. ***$P < 0.001$. ns not significant. **c** Plasma viral loads after productive infection with SIVmac239 in the 3 animal groups: K11 (red), 5L7-LS (blue), and control (black). The threshold of the assay employed was 15 SIV RNA copies/mL. The plasma viral concentrations for each animal were indicated by different symbols. **d** Comparison of peak plasma viral loads among the 3 groups of infected animals based on two-tailed Mann–Whitney U test. ns not significant ($p > 0.05$). **e** Comparison of set point viral load among three groups of infected animals based on two-tailed Mann–Whitney U test. Set point viral load was the geometric mean of all values from each animal between weeks 10 and 20. **$P = 0.0043$. **f** Plasma nAb $ID_{50}$ titers and geometric mean of $ID_{50}$ titer 7 days prior to infection in the K11 group using the SIVmac239 pseudovirus assay. N/A not applicable. **g** ADCC activity was measured for rhesus mAbs using SIVmac239 virus on CEM.CCR5 target cells and macaque NK effector cells. CD4-IgG2 served as positive control and an anti-dengue mAb Den3 as a negative control. Data are presented as mean values ± SD, and error bars are from two technical replicates.

to determine whether combining antibody and T cell responses can enhance protection. This investigation would be most readily conducted in NHPs, but that requires deploying both arms of immunity against either SHIV or SIV. Although both virus models are informative with different strengths and weaknesses, we are focused here on SIV given the promising preclinical T cell vaccine studies with CMV vectors and considering that SIV is considered a more robust model of HIV and more reflective of HIV genotype diversity. Accordingly, we report here the generation of appropriate humoral reagents, namely SIV-specific nAbs and an SIV trimer immunogen, and describe the generation and characterization of both.

We isolated 12 nAbs against SIVmac239, characterized one (K11) in some detail, and showed that it binds to a glycan hole on gp120 and that it can protect macaques against SIVmac239 IV challenge. While IV

challenge is not the most frequent route of HIV transmission, it is a more stringent challenge compared to intravaginal and intrarectal routes. The serum nAb titers required for protection against this route were comparable to those noted for protection against HIV in humans and SHIV in macaques[15,52]. Thus, at least from a nAb standpoint, it does not appear that SIVmac239 is more resistant per se to neutralization or requires higher nAb titers for protection than HIV or SHIV. Rather, in infected macaques, the virus tends to elicit nAb responses that are of low neutralization potency, rare, or delayed compared with many HIV and SHIV infections, reflecting the poor immunogenicity of nAb epitopes on SIVmac239 Env. Notwithstanding, the nAb K11 is now available as a passive reagent to be combined with T cell responses to investigate the synergy of humoral and cellular protection against the rigorous SIVmac239 challenge.

We generated a stabilized recombinant trimer of SIVmac239 and solved its structure in complex with nAb K11 by cryo-EM. This is the first high-resolution structure of a soluble macaque SIV trimer. The trimer was shown to closely resemble trimers of HIV and of chimpanzee SIV but with distinct features, notably a V4 loop and an extended V1 loop that fold back on top of the gp120 core; conformations made possible by differences in the glycan shield relative to HIV and SIVcpz. Although SIVmac239 contains two additional glycans on its extended V1 loop, our computational modeling revealed it is still one of the most poorly shielded regions of the trimer. This is consistent with the prevalence of antibodies targeting this region, such as FZ019.2. The absence of antibodies targeting the rest of the highly exposed V1 loop in all the polyclonal sera tested is perplexing, especially relative to the abundance of V4/V5-targeting antibodies, seeing how V5 is significantly more shielded by glycans. The functional significance of this exposed extended V1 loop is unclear, though it could be acting to protect the more conserved gp120 core, which is shielded by glycans on SIVcpz and HIV Env, but not on SIVmac. A HTAM pipeline was used to generate a large ensemble of fully glycosylated SIVmac239 trimer structures and show that the SIVmac239 trimer is more densely packed with N-linked glycans than the prototypic BG505 HIV trimer. This higher shielding may be associated with the relatively low levels of nAb responses to the SIVmac239 trimer in natural infection[13]. Further, the abundance of antibody responses to degraded forms of Env, likely related to the poor stability of the trimer, may also play a role. Such non-nAb responses to SIVmac239 infection parallel those seen in HIV infection that presumably arises from strong antibody responses to nonfunctional forms of Env such as gp41 stumps and V3 loop[39]. The V4/V5 region, however, is a neutralizing epitope for some HIV strains but non-neutralizing for SIVmac239. This observation may be related to more glycan shielding on the SIVmac239 V4/V5 loop than on many HIV strains. The availability of the recombinant trimer will now, however, permit exploration of the molecular origins of low native trimer immunogenicity in macaques. Additionally, the high-resolution structure also enables rational engineering of the trimer to improve stability and potentially immunogenicity. Indeed, previous studies suggest that the removal of one or more glycans from SIVmac239 Env can enhance nAb responses to wild-type virus induced by infection with the glycan-deficient virus[58,59].

Previous studies report on the possible contributions of non-neutralizing antibodies[60,61] elicited by SIVDeltaNef constructs to SIVmac239 protection. Our study reports protection against SIVmac239 via passive transfer of a neutralizing antibody and failure to protect with the transfer of a non-neutralizing antibody, although we note that this study is focused on monoclonal antibodies and not polyclonal responses elicited by vaccination.

Overall, our studies represent a notable advance in our understanding of the interaction of the humoral immune system with the prototypic SIVmac239 and pave the way for exploring synergies between humoral and cellular immune systems in the SIV model that will directly inform HIV vaccine design strategies.

## Methods

### Rhesus macaques

The 18 Indian rhesus macaques (Macaca mulatta) used in the current study (11 males and 7 females) were housed at the Wisconsin National Primate Research Center (WNPRC), cared for in accordance with Weatherall report guidelines, and the principles described in the National Research Council Guide for the Care and Use of Laboratory Animals. All procedures were previously approved by the University of Wisconsin Graduate School Animal Care and Use Committee (animal welfare assurance number 16-00239 [A3368-01]; protocol number G005248). Rhesus macaques were separated as follows: Group 1 (5L7-LS-infused RMs, $n = 6$), Group 2 (K11-infused RMs, $n = 6$), Group 3 (untreated RMs, $n = 6$). Both 5L7-LS and K11 were prepared in saline

bags and administered intravenously into each animal at 60 mg/kg before the first challenge. Three weeks after, a second infusion was given to all Group 1 and Group 2 animals.

### Cell lines

Irradiated 3T3msCD40L (ATCC) cells were used in a single B cell culture assay. TZM-bl cells (NIH AIDS Reagents Program) were used in pseudovirus neutralization assay. Human HEK 293 T cells (ATCC) were used for pseudovirus production. HEK 293 S cells (ATCC) were used for the generation of pseudovirus expressing Man5 glycans. FreeStyle HEK 293 cells (ThermoFisher) were used for recombinant Env protein production. Expi293F cells ((ThermoFisher) were used for monoclonal antibody production. CEM.NKR-CCR5-sLTR-Luc cells[62] were used for ADCC assay.

### Recombinant Env protein purification

SIVmac239 SOSIP.664-His, SIVmac239 gp140 FT, SIVmac239 gp120, and other Env mutant constructs were transiently expressed in FreeStyle 293F cells (ThermoFisher) as described previously[27,34]. SIVmac239 SOSIP.664 supernatants were purified by affinity chromatography using a PGT145 column. Supernatants of SIVmac239 gp140 FT, SIVmac239 gp120, and other mutant constructs were purified by an agarose-bound Galanthus Nivalis Lectin (GNL) column (VectorLab). Affinity-purified proteins were then purified by size exclusion chromatography using a HiLoad 26/600 Superdex 200 pg column (GE Healthcare).

### Isolation of SIVmac239 SOSIP.664-specific memory B cells by flow cytometry

Cryopreserved PBMCs were thawed, washed, and stained with an antibody cocktail (1:100 dilution) of CD3 (clone SP34-2, BD Biosciences), CD4 (clone OKT4, Biolegend), CD8 (clone RPA-T8, BD Biosciences), CD14 (clone M5E2, BD Biosciences), CD20 (clone 2H7, Biolegend), IgM (MHM-88, Biolegend), IgG (clone G18-145, BD Biosciences) and fluorescently labeled biotinylated SIVmac239 SOSIP.664 Env at room temperature for 20 min in the dark. SIVmac239 Env trimer probes were labeled with two different fluorophores, from which SIVmac239 Env dual positive memory B cells (CD3⁻CD4⁻CD8⁻CD14⁻CD20⁺IgM⁻IgG⁺SIVmac239 SOSIP²⁺) were analyzed with BD FACSMelody or BD FACSFusion, and single-cell sorted, cultured, expanded in 384-well plates as described previously[28,29]. Briefly, sorted B cells were cultured with Iscove's modified Dulbecco's medium (IMDM) with GlutaMAX (Gibco) supplemented with 10% heat-inactivated fetal bovine serum (FBS), 1× MycoZap Plus-PR (Lonza), 100 U/mL human IL-2 (Roche), 50 ng/mL human IL-21 (Invitrogen), 50 ng/mL human IL-4 (Miltenyi), 0.1 μg/mL anti-rhesus IgG (H + L) (BioRad), and irradiated 3T3msCD40L feeder cells. Flow cytometric data were subsequently analyzed using FlowJo (v10.7.1).

### SIV pseudovirus production

The panel of SIV pseudovirus constructs was kind of gifts from Mario Roederer at NIH, David Montefiori at Duke University, and Cynthia Derdeyn at Emory University. SIV pseudovirus Env construct was cotransfected with *env*-deficient backbone plasmid (pSG3ΔEnv) in a 1:2 ratio with transfection reagent FuGENE 6 (Promega) in HEK 293 T cells according to the manufacturer's instructions. To investigate the role of glycosylation for FZ019.2 antibody neutralization, two approaches were used: glycosidase inhibitors and the mutant GnT1-deficient HEK 293 S cell line (GnT1⁻/⁻)[63]. Glycosidase inhibitors were added at the time of transfection and were used at the following concentrations: 25 μM kifunensine and 20 μM swainsonine. Mutant virus Env plasmid was generated by Quikchange site-directed mutagenesis kit (Agilent) following the manufacturer's instructions. After 72 h of transfection, supernatants containing viruses were harvested

and sterile filtered (0.22 μm) (EMD Millipore) and frozen at −80 °C for long-term storage.

## Neutralization screening

After 2 weeks of culture, supernatants were carefully collected and screened in SIVmac239 pseudovirus micro-neutralization assay. Twenty microliters of culture supernatants were incubated with SIVmac239 pseudovirus in sterile 384-well white plates (Greiner Bio-One) for 1 h at 37 °C. Then 20 μL of TZM-bl cells with diluted DEAE-dextran were added to each well at a concentration of 0.2 million cells/mL and incubated for 48 h at 37 °C. After that, culture media were removed, cells were lysed, and luciferase activity was measured by adding BrightGlo (Promega) according to the manufacturer's instructions. Sorted wells with detectable IgG secretion and over 50% neutralization activity were selected to amplify antibody genes.

## RT-PCR, Ig amplification, cloning, and antibody production

After removing culture supernatants, cells in 384-well plates were lysed, and mRNA was extracted using TurboCapture 384 mRNA kit (QIAGEN, 72271) following the manufacturer's instructions. mRNA plates were stored at −80 °C. Single-cell RNA from neutralization positive wells were reverse transcribed as described previously[64]. Subsequently, nested PCR reactions of Ig heavy chain and light chain variable regions were performed in 25 μL volume with 2 μL cDNA transcript. For the first round of PCR in 25 μL reaction, 2.5 μL of 10X PCR buffer (QIAGEN), 0.25 μL of HotStarTaq Plus DNA Polymerase (QIAGEN), 0.5 μL of dNTP (10 mM) (ThermoFisher), 0.25 μL of MgCl$_2$ (25 μM), 0.25 μL of forwarding primer mixture (50 μM each primer), 0.25 μL of reverse primer mixture (25 μM each primer), 19 μL of DEPC-treated water (ThermoFisher), 2 μL of gene transcripts. Rhesus primers and PCR program were described previously[34]. Two microliters of PCR1 products were amplified by nested PCR using 5 μL of 5X Phusion HF buffer (Thermo Fisher), 0.24 μL of Phusion HF DNA polymerase (2 U/μL, Thermo Fisher), 0.5 μL of dNTP, 0.25 μL of forwarding primers and 0.25 μL of reverse primers with Gibson adaptor sequences, 0.75 μL of 2X MgCl$_2$, using the following PCR program: 30 s at 98 °C, followed by 35 cycles of 10 s at 98 °C, 40 s at 72 °C, and a final extension for 5 min. Amplified PCR products were analyzed with 2% 96 E-gel (ThermoFisher). Antibodies with recovered and paired heavy chains and light chains were cloned into the corresponding Igγ1, Igκ, Igλ expression vectors[64]. Cloned rhesus HC and LC variable regions were sequenced (Genewiz) and subsequently analyzed using the rhesus germline database[31].

Antibody HC and LC constructs were transiently expressed with the Expi293 Expression System (ThermoFisher). HC and LC plasmids were cotransfected at a 1:2.5 ratio with transfection reagent FectoPRO (Polyplus) in Expi293 cells according to the manufacturer's instructions. Twenty-four hours post-transfection, cells were fed with 300 mM valproic acid and 40% glucose (Gibco). After 4–5 days of transfection, cell supernatants were harvested and sterile filtered (0.22 μm). Antibody was purified by Protein A Sepharose (GE Healthcare) as described previously[65].

## TZM-bl neutralization assay

Serially diluted serum or antibody sample was incubated with SIV pseudovirus in half-area 96-well white plates using Dulbecco's Modified Eagle Medium (DMEM) (Gibco) supplemented with 10% FBS, 2 mM L-glutamine (Gibco), 100 U/mL Penicillin/Streptomycin (Gibco). After 1 h incubation at 37 °C, TZM-bl cells with diluted DEAE-dextran were added onto the plates at 10,000 cells/well. Final antibody concentrations for the dilution series were calculated based on the total volume of the assay (antibody + virus supernatant + TZM-bl cells). For testing antibody neutralization against the replication-competent virus, serially diluted mAbs were incubated with replication-competent SIVmac239 for 1 h at 37 °C, then transferred onto TZM-bl

cells containing 1 μM Indinavir (Sigma) in half-area 96-well plates (Corning). After 48 h of incubation, culture supernatants were removed, and cells were lysed in luciferase lysis buffer (25 mM Gly-Gly pH 7.8, 15 mM MgSO$_4$, 4 mM EGTA, 1% Triton X-100). Luciferase activity was measured by adding BrightGlo (Promega) according to the manufacturer's instructions. Assays were tested in duplicate wells and independently repeated at least twice. Neutralization IC$_{50}$ or ID$_{50}$ titers were calculated using "One-Site Fit LogIC$_{50}$" regression in GraphPad Prism 8.0. The maximum percentage of neutralization was defined as the maximum % neutralization observed at the highest mAb concentrations tested.

## Recombinant protein binding ELISA

6x His tag monoclonal antibody (Invitrogen) or streptavidin (Jackson ImmunoResearch) was coated at 2 μg/mL in PBS onto half-area 96-well high binding plates (Corning) overnight at 4 °C. After washing, plates were blocked with PBS/3% BSA for 1 h at RT. His-tagged recombinant SIVmac239 Env protein was added at 1 μg/mL onto plates precoated with 6× His tag antibody while biotinylated SIVmac239 Env protein was added at 1 μg/mL onto plates precoated with streptavidin. Protein was captured and incubated for 1 h at RT. After washing, serially diluted mAb in PBS/1% BSA was added in duplicates into wells and incubated for 1 h at RT. After washing, 1:1000 diluted alkaline phosphatase-conjugated goat anti-human IgG Fcγ antibody (Jackson ImmunoResearch) was added into wells and incubated at RT for 1 h. After the final wash, phosphatase substrate (Sigma–Aldrich) was added to the wells. Absorption was measured at 405 nm. Nonlinear regression curves were analyzed using Prism 8 software to calculate EC$_{50}$ values.

## Competition ELISA

6x His tag monoclonal antibody (Invitrogen) was coated at 2 μg/mL in PBS onto half-area 96-well high binding plates (Corning) overnight at 4 °C. After washing, plates were blocked with PBS/3% BSA for at least 1 h at RT. His-tagged SIVmac239 SOSIP.664 protein was added into wells at 1 μg/mL and incubated for 1 h at RT. After washing, serially diluted mAbs (started at 50 μg/mL) or sera (started at 1:50 dilution) in PBS/1% BSA were added into wells and incubated for 30 min at RT. After removing liquid, biotinylated mAb was added at a concentration corresponding to EC$_{70}$ and incubated for 45–60 min. After washing, alkaline phosphatase-conjugated streptavidin (Jackson ImmunoResearch) was diluted at 1:1000 in PBS/1% BSA and added to the wells at RT for 1 h. After the final wash, phosphatase substrate (Sigma–Aldrich) was added to the wells. Absorption was measured at 405 nm.

## Negative-stain Electron Microscopy (nsEM) sample preparation, imaging, and data processing

The monoclonal antibody was digested into Fab using papain resin and purified by protein A beads (ThermoFisher) according to the manufacturer's instructions. Rhesus sera or plasma antibodies were first purified by affinity chromatography using Protein A Sepharose (GE Healthcare) according to the manufacturer's instructions. Then polyclonal serum antibodies were digested into Fab using papain resin. Purified SIVmac239 trimer was then mixed with purified Fab or IgG and incubated overnight, followed by purification with size exclusion chromatography. Antibodies were added at a 6× molar excess to trimer for monoclonal complexes and ~30× molar excess for polyclonal complexes. Purified complexes were then concentrated to ~0.01 mg/mL and applied to plasma-cleaned carbon grids, followed by two rounds of staining with uranyl formate (UF). Imaging was performed on an FEI Tecnai Spirit T12 transmission electron microscope operating at 120 keV and equipped with an Eagle 4 K CCD. A magnification of 52,000× was used, resulting in a physical pixel size of 2.05 Å. Automated data collection was performed using the Leginon software package[66]. All single-particle data processing steps were performed using Relion/3.0[67]. Figures were generated using UCSF Chimera[68] by

aligning representative 3D reconstructions to a ligand-free molecular surface model of SIVmac239, followed by segmentation of Fab regions using Segger[69]. For clarity, figures only display one Fab density per epitope.

## Cryo-EM sample preparation and imaging

Purified K11 IgG was mixed with purified SIVmav239 trimer then complexes were immediately purified via SEC and concentrated to ~1 mg/mL for cryo-EM preparation. The concentrated sample was mixed with 0.5 μL of 0.04 mM lauryl maltose neopentyl glycol (LMNG; Anatrace) to a final concentration of 0.005 mM, and 4 μL of this solution was applied to plasma-cleaned 1.2/1.3 C-Flat holey carbon grids (Electron Microscopy Sciences) using a Vitrobot mark IV (Thermo Fisher Scientific) with a 7 sec blot time, 0 blot force, and wait time of 0 sec. Prepared grids were then stored in liquid nitrogen until they were transferred to a microscope for imaging. A table of detailed imaging conditions is presented in Table S2. Data was collected with Leginon automated microscopy software[66] on an FEI Titan Krios operating at 300 keV (Thermo Fisher Scientific) equipped with a K2 Summit direct electron detector (Gatan) operated in counting mode.

## Cryo-EM data processing

All movie micrographs were aligned and dose-weighted using MotionCor2[70], and CTF parameters were estimated with GCTF[71]. Single-particle processing was carried out using CryoSparc2[72]. The following general workflow was used for single-particle data processing. After frame alignment, dose-weighting, and CTF estimation, micrographs were sorted based on CTF fit parameters, and particle picking was performed first using a gaussian blob template on a subset of micrographs. These particles were then extracted, aligned, and classified in 2D, and the class averages were used for template picking of the full dataset. Picked particles were subjected to one or two rounds of 2D classification followed by subset selection. One round of ab initio classification was carried out, followed by subset selection of all classes containing well-refined trimers. Next, 3D-autorefinement was performed with per-particle CTF estimation and global CTF refinement using C3 symmetry. It was observed that the map density for gp41 was poorly resolved, so we performed another round of 3D classification using the 3D variability algorithm implemented in CryoSparc2[73] followed by clustering into 6 classes. Analysis of the variability along principal components 1–3 revealed an asymmetric "scissoring" and/or twisting motion of the trimer that gave rise to significant displacement of the HR1 and HR2 helices (Supplementary Movie 1). Clusters representing the extreme ends of these variability modes were then pooled and refined without imposing symmetry. This asymmetric map displayed fully resolved gp41 HR1 and HR2 helices in one of the three protomers (Supplementary Fig. 4j).

## Model building and figure preparation

Model building was initiated by preparing a monomeric SIVmac239 homology model with SWISS-MODEL[74] using MT145K (PDB:6OHY) as a template. A homology model of K11 was generated using SAbPred[75]. Preliminary Env and Fab models were then fit into the asymmetrically refined cryo-EM map using the protomer with the best-resolved gp41 domain and combined into a single PDB file using UCSF Chimera[68]. The novel V1 loop conformation was built de novo along with all glycans using COOT[76], and gp41 helices were adjusted using rigid body docking. Fully glycosylated models were then refined in Rosetta, asking for ~300 models[77]. All models were validated using MolProbity[78] and EMRinger[79] and the model with the best-combined score was selected. The model was then checked and adjusted manually in COOT and refined with Rosetta, if necessary. The model from the protomer with the most well-resolved gp41 from the asymmetric reconstruction was then fit and relaxed into the C3 symmetric map and refined with Rosetta. Final models have then scored again with MolProbity and

EMRinger, while glycan structures were further validated with Privateer[80]. Figures were prepared with either UCSF Chimera or ChimeraX[81]. Hydrogen bonds were calculated and displayed with UCSF ChimeraX. Volume segmentation was performed with Segger[82] as implemented in UCSF ChimeraX. The 3D variability movies were generated with UCSF Chimera and edited in Blender.

## Modeling to quantify the extent of glycan shielding of the SIV mac239 Env surface

We first generated an ensemble of non-glycosylated protein models using Rosetta multi-cycle refinement[77,83] with our 3.4Å-resolution cryo-EM map and chose the top 10 models based on a combined EMRinger[79] and MolProbity[78] score to serve as scaffolds. We then built 100 possible glycosylated conformations onto each scaffold, relaxed them over several steps of conjugate gradient energy minimization, and simulated annealing, using our HTAM pipeline[46]. Glycosylation at each PNGS was modeled based on the site-specific predominant glycan type as determined by MS. These 1000 different conformations formed the final glycoprotein ensemble. From this ensemble, we calculated the root mean squared fluctuation of each glycan, averaged over the carbon and oxygen atoms of the β-mannose ring. The per-residue glycan encounter factor (GEF) score[49], which quantifies the shielding effect over the protein surface, was calculated as the geometric mean of the probability that surfaces orthogonal vectors (normal and planar) located at each residue would encounter glycan heavy atoms. For comparison, we also performed the same calculations on an equally sized ensemble of fully glycosylated BG505 SOSIP.664 models.

## Site-specific glycan analysis

Three aliquots of SIVmac239 were denatured for 1 h in 50 mM Tris/HCl, pH 8.0 containing 6 M of urea, and 5 mM dithiothreitol (DTT). Next, Env proteins were reduced and alkylated by adding 20 mM iodoacetamide (IAA) and incubated for 1 h in the dark, followed by a 1 h incubation with 20 mM DTT to eliminate residual IAA. The alkylated Env proteins were buffer-exchanged into 50 mM Tris/HCl, pH 8.0 using Vivaspin columns (3 kDa), and two of the aliquots were digested separately overnight using trypsin, chymotrypsin (Mass Spectrometry Grade, Promega) or alpha lytic protease (Sigma–Aldrich) at a ratio of 1:30 (w/w). The next day, the peptides were dried and extracted using C18 Zip-tip (MerckMilipore). The peptides were dried again, resuspended in 0.1% formic acid, and analyzed by nanoLC-ESI MS with an Ultimate 3000 HPLC (Thermo Fisher Scientific) system coupled to an Orbitrap Eclipse mass spectrometer (Thermo Fisher Scientific) using stepped higher energy collision-induced dissociation (HCD) fragmentation. Peptides were separated using an EasySpray PepMap RSLC C18 column (75 μm × 75 cm). A trapping column (PepMap 100 C18 3 μM 75 μM × 2 cm) was used in line with the LC prior to separation with the analytical column. The LC conditions were as follows: 280 min linear gradient consisting of 4-32% acetonitrile in 0.1% formic acid over 260 min followed by 20 min of alternating 76% acetonitrile in 0.1% formic acid and 4% Acn in 0.1% formic acid, used to ensure all the sample had eluted from the column. The flow rate was set to 300 nL/min. The spray voltage was set to 2.5 kV, and the temperature of the heated capillary was set to 40 °C. The ion transfer tube temperature was set to 275 °C. The scan range was 375–1500 $m/z$. The stepped HCD collision energy was set to 15, 25, and 45%, and the MS2 for each energy was combined. Precursor and fragment detection were performed using an Orbitrap at a resolution MS1 = 120,000. MS2 = 30,000. The AGC target for MS1 was set to standard and injection time set to auto, which involves the system setting the two parameters to maximize sensitivity while maintaining cycle time. Full LC and MS methodology can be extracted from the appropriate Raw file using XCalibur FreeStyle software or upon request.

Glycopeptide fragmentation data were extracted from the raw file using Byos (Version 3.5; Protein Metrics Inc.). The glycopeptide

fragmentation data were evaluated manually for each glycopeptide; the peptide was scored as true-positive when the correct b and y fragment ions were observed along with oxonium ions corresponding to the glycan identified. The MS data were searched using the Protein Metrics 305 N-glycan library with sulfated glycans added manually. The relative amounts of each glycan at each site as well as the unoccupied proportion were determined by comparing the extracted chromatographic areas for different genotypes with an identical peptide sequence. All charge states for a single glycopeptide were summed. The precursor mass tolerance was set at 4 ppm and 10 ppm for fragments. A 1% false discovery rate (FDR) was applied. The relative amounts of each glycan at each site as well as the unoccupied proportion were determined by comparing the extracted ion chromatographic areas for different glycopeptides with an identical peptide sequence. Glycans were categorized according to the composition detected.

HexNAc(2)Hex(10 + ) was defined as M9Glc, HexNAc(2)Hex(9 − 5) was classified as M9 to M3. Any of these structures containing fucose were categorized as FM (fucosylated mannose). HexNAc(3)Hex(5 − 6)X was classified as Hybrid with HexNAc(3)Hex(5-6)Fuc(1)X classified as Fhybrid. Complex-type glycans were classified according to the number of HexNAc subunits and the presence or absence of fucosylation. As this fragmentation method does not provide linkage information, compositional isomers are grouped so for example, a triantennary glycan contains HexNAc 5 but so do biantennary glycans with a bisect. Core glycans refer to truncated structures smaller than M3. M9glc- M4 were classified as oligomannose-type glycans. Glycans containing at least one sialic acid or one sulfate group were categorized as NeuAc and sulfated, respectively.

## SIVmac239 Env peptide ELISA screening

Biotinylated SIVmac239 Env 15-mer peptides were synthesized at GenScript. Overlapping peptide sequences were shown in Supplementary Table 2. Streptavidin (Jackson ImmunoResearch) was coated at 2 μg/mL in PBS onto half-area 96-well high binding plates (Corning) overnight at 4 °C. After washing, plates were blocked with PBS/3% BSA for 1 h at RT. After washing, the peptide library was diluted to 50 μg/mL in PBS and added onto individual wells of ELISA plates, and incubated for 1 h. After washing, 1:50 diluted rhesus sera or 10 μg/mL mAbs were added to each plate and incubated for 1 h at RT. After washing, 1:1000 diluted alkaline phosphatase-conjugated goat anti-human IgG Fcγ antibody (Jackson ImmunoResearch) was added into wells and incubated at RT for 1 h. After the final wash, phosphatase substrate (Sigma–Aldrich) was added to the wells. Absorption was measured at 405 nm.

## ADCC assay

CEM.NKR-CCR5-sLTR-Luc cells[62], which express luciferase upon infection, were infected with SIVmac239. After two days of infection, serially diluted mAbs were added to the target cells and incubated for 15 min. Primary human NK cells were added for an approximate 10 to 1 effector to target ratio. Cells were incubated overnight. Luciferase activity was measured using BrightGlo (Promega) as described above. Uninfected or infected cells incubated with NK cells in the absence of antibodies were used to determine the background and maximal luciferase activity, respectively. The dose-dependent loss of luciferase activity represents the antibody-dependent killing of productively infected target cells.

## SIVmac239 challenge

The growth and titer of the SIVmac239 challenge stock were performed as described previously[84], and the same challenge stock[5,9] was used in this study. All animals were subjected to repeated intravenous inoculations of $10^{5.5}$ rhesus monkeys per mL (AID$_{50}$) of SIVmac239 (Genbank ID M33262). The challenge virus was administered in 1 mL of PBS. The intravenous challenge occurred weekly. Plasma samples were collected on days 7 and 10 post-challenge to assess for viral loads. Animals with detectable viral loads at either of those time points were no longer challenged.

## Viral load measurement

Viral loads were measured using 0.5 mL of plasma, as previously described[85]. Total RNA was extracted from each sample using QIAgen DSP virus/pathogen Midi kits (QIAGEN) on a QIASymphonyXP laboratory automation instrument platform. A two-step RT-PCR reaction was performed using a random primed reverse transcription reaction, followed by 45 PCR cycles. Each sample had six replicates. The following primers and probes were used: forward primer: SGAG21: 5′-GTCTGCGTCAT(dP)TGGTGCA TTC-3′; reverse primer SGAG22: 5′-CACTAG(dK)TGTCTCTGCACTAT(dP)TGTTTTG-3′; probe: PSGAG23: 5′-FAM528.

CTTC(dP)TCAGT(dK)TGTTTCACTTTCTCTTCTGCG-BHQ1- 3′. As performed, the threshold sensitivity of the assay was 15 SIV RNA copies/mL plasma.

## Plasma antibody ELISA

Plasma antibody levels of K11 and C9-tagged 5L7 were measured by ELISA using plates coated overnight at 2 μg/mL with either purified SIVmac239 gp140 FT or mouse anti-rhodopsin (C9) monoclonal Ab (EMD Millipore, 6B9575), respectively. Following overnight coating, plates were washed with 1× PBS-Tween20 and blocked with 5% nonfat dry milk in PBS at 37 °C for 1 h. After blocking, plates were washed, and 50 μL of serially diluted serum and purified K11 or 5L7 was added to the corresponding wells. Plates were then washed before adding goat anti-human IgG-HRP (SouthernBiotech, 2045-05) diluted 1:10,000 to all wells and incubating for 1 h at 37 °C. After a final wash, plates were developed using 3,3′,5,5′-Tetramethylbenzidine substrate (SouthernBiotech, 0410-01), and the reaction was stopped using TMB stop solution (SouthernBiotech, 0412-01) before reading absorbance at 450 nm.

## Statistical analysis

General quantification and standard deviations were calculated between technical replicates using GraphPad Prism 8.0. The Kaplan–Meier method and log-rank test were used to determine whether an infusion of K11 affected the acquisition of SIVmac239. The time to detectable infection was analyzed using Kaplan–Meier method, and the differences between K11 group and the control group, or between 5L7-LS group and the control group were evaluated using the log-rank test. The Mann–Whitney test was used to compare the peak viral load and the set point viral load of infected animals in K11 or 5L7-LS group with the control group.

## Reporting summary

Further information on research design is available in the Nature Research Reporting Summary linked to this article.

# Data availability

The data that support the findings of this study are available from the corresponding authors upon request. Antibody sequences have been deposited to Genbank (accession codes ON526845-ON526868). Cryo-EM and nsEM structures and refined atomic models have been deposited to the Electron Microscopy Data Bank under accession codes EMD-25621, EMD-25676, EMD-25623,EMD-25624,EMD-25625, EMD-25626, EMD-25627, EMD-25628, EMD-25629, EMD-25630, EMD-25631, and EMD-25632, and the Protein Data Bank under accession codes 7T2P and 7T4G. Glycopeptide LC-MS Raw files can be accessed at ftp://massive.ucsd.edu/MSV000089876/. Antibody plasmids and recombinant SOSIP plasmids in this study are available with a material transfer agreement. Source data are provided with this paper.

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

## Acknowledgements
We thank Rosemarie Mason and Mario Roederer at NIH, David Montefiori at Duke University, and Cynthia Derdeyn at Emory University for providing the SIV pseudovirus envelope plasmids. We thank Louis Picker at Oregon Health & Science University for providing the non-neutralizing SIVmac239-infected macaque plasma samples. We thank Matthias Pauthner for testing the plasma nAb titers from previous infected rhesus samples and the staff of the Quantitative Molecular Diagnostics Core of the AIDS and Cancer Virus Program, Frederick National Laboratory for technical assistance with viral load measurements. This work is supported by the National Institute of Allergy and Infectious Diseases (NIAID) Consortium for HIV/AIDS Vaccine Development (CHAVD; UM1AI144462) (M.C., A.B.W., D.R.B., and D.S.), the Bill and Melinda Gates Foundation through the Collaboration for AIDS Vaccine Discovery (INV-008352/OPP1153692 and OPP1196345/INV-008813) (D.S., D.R.B., A.B.W., and R.A.), and the NIH R61 AI161818 (R.A.); in part by NIH Grant R61 AI161818 (R.A.), NIH RO1 AI 52056-20 (D.I.W.), and Federal funds from the National Cancer Institute, National Institutes of Health, under Contract Nos. HHSN261201500003I and 75N91019D00024 (J.D.L.). This work is also supported in part by the Office of The Director, National Institutes of Health under Award Number P51OD011106 to the Wisconsin National Primate Research Center, University of Wisconsin-Madison. This research was conducted in part at a facility constructed with support from Research Facilities Improvement Program grant numbers RR15459-01 and RR020141-01.

## Author contributions
F.Z., D.R.B., and D.S. conceived and designed the study. F.Z. and A.B. performed sorting, antibody cloning, and sequencing experiments. F.Z., A.B., S.B., and O.L. expressed and purified the monoclonal antibodies. N.P., J.L., T.V., F.L., B.R., and M.R. expressed and purified 5L7-LS antibody. F.Z., A.B., and S.B. characterized monoclonal antibodies in functional assays. F.Z. and A.B. expressed the recombinant SIVmac239 gp120 delta V1V2, delta V3, and gp140 FT proteins. G.S., P.Y., and R.A. expressed the recombinant SIVmac239 SOSIP.664 protein for cryo-EM and nsEM. F.Z., A.B., and S.B. produced pseudovirus and performed TZM-bl assay. Z.T.B. performed cryo-EM experiments, data processing, and model building. Z.T.B., W.L., L.S., and G.O. performed nsEM experiments and Z.T.B. and W.L. processed the data. J.D.A. performed LC-MS analysis; J.D.A. and M.C. analyzed and interpreted data. R.C.D. provided the SIVmac239 challenge stock. R.N. provided the neutralization data against challenge stock and the ADCC data. E.G.R., D.B., N.P., R.C.D., and D.I.W. planned the animal challenge studies. K.L.W. and E.G.R. performed the antibody infusion and SIVmac239 challenge study. K.L.W. and E.G.R. collected the plasma samples, and J.D.L. measured viral load. N.P. and D.I.W. analyzed peak viral load and measured rhesus serum antibody concentrations. S.C. performed computational modeling of the glycan shield and S.C. and Z.T.B. analyzed and interpreted the modeling data. F.Z., Z.T.B., A.B.W., D.R.B., and D.S. wrote the manuscript and all authors reviewed and edited the manuscript.

## Competing interests
The authors declare no competing interests.

## Additional information

[1]Department of Immunology and Microbiology, The Scripps Research Institute, La Jolla, CA 92037, USA. [2]IAVI Neutralizing Antibody Center, The Scripps Research Institute, La Jolla, CA 92037, USA. [3]Consortium for HIV/AIDS Vaccine Development (CHAVD), The Scripps Research Institute, La Jolla, CA 92037, USA. [4]Department of Integrative Structural and Computational Biology, The Scripps Research Institute, La Jolla, CA 92037, USA. [5]Department of Pathology, George Washington University, Washington, DC 20037, USA. [6]School of Biological Sciences, University of Southampton, Southampton SO17 1BJ, UK. [7]Theoretical Biology and Biophysics Group, Los Alamos National Laboratory, Los Alamos, NM 87545, USA. [8]IAVI, New York, NY 10004, USA. [9]Wisconsin National Primate Research Center, University of Wisconsin-Madison, Madison, WI 53715, USA. [10]AIDS and Cancer Virus Program, Frederick National Laboratory for Cancer Research, Frederick, MD 21701, USA. [11]Department of Pathology, Miller School of Medicine, University of Miami, Miami, FL 33136, USA. [12]Ragon Institute of Massachusetts General Hospital, Massachusetts Institute of Technology, and Harvard University, Cambridge, MA 02139, USA. [13]These authors contributed equally: Fangzhu Zhao, Zachary T. Berndsen, Nuria Pedreño-Lopez. ✉e-mail: andrabi@scripps.edu; andrew@scripps.edu; burton@scripps.edu; dsok@iavi.org

