## [Peer Review File · Nature Communications]

Reviewer comments, first round -

Reviewer #1 (Remarks to the Author):

Comments:

The authors provide a very inclusive study of antibody mediated protection in the SIV model. Their studies of SIVmac239 infection of macaques is the best model for human HIV type-1 infection. Their approach using cryo-EM structures of the SIV envelope trimer appears highly likely to provide insights for vaccine design for the HIV-1 envelope, and surely further critical understanding of broadly neutralizing antibodies (bnAbs) in humans that have possible clinical application. Their major goal appears to address an important question directed to future HIV-1 virology. That is for their model, the use of nAb K11 as a passive reagent to be combined with T cell responses to investigate synergy of humoral and cellular protection against a SIVmac239 challenge.

The isolation and characterization of 12 potent SIVmac239 monoclonal nAbs from chronically infected macaques was well conceived and clearly presented. The identification of K11 neutralizing antibody and its subsequent use in the cryo-EM of the gp140 structure was probably the major accomplishment in this paper. The authors should again cross-check the data including peptide sequences in all figures and their legends as mentioned below. Is the identification of 7 mAbs to non-neutralizing SIVmac239 epitopes that are immunodominant related to the same situation in humans, potentially a discussion point for new readers not familiar with antibodies identified in early and late infected humans by HIV-1?

The cryo-EM data of the SIVmac239 env trimer in complex with the nAb K11 was a major accomplishment, particularly at an overall 3.4 Å resolution. The presentation of the structural data was excellent. The extensive mapping of the glycans onto the trimer, a major effort by itself, was also well presented. The authors present a compelling argument for glycan content of the trimer and its relationship to low immunogenicity.

The studies which show protection of rhesus macaques by nAb from repeated SIVmac239 challenge highly suggest humoral immunity plays an important role in protection. The literature cited was excellent and inclusive.

Minor points:

The manuscript is littered with a lot of typos maybe because of contributions from many investigators from various areas. Some are listed below. The authors should be sure that the sequences of the peptides listed in all Figures and Tables are correct. Correlation of what is the text and that information that is in the figures and legends should also be checked again.

Line 179, space between (Fig. 2c,dand...) space between d/and.

Line 189, Fig. 2d should read 2e

Line 1021, Page 35: "gaussian blot template" should be "gaussian plot template"

Reviewer #2 (Remarks to the Author):

In this paper titled "Molecular insights into antibody-mediated protection against the prototypic simian immunodeficiency virus", Zhao et al. have isolated 12 potent nAbs against SIVmac239 virus from three chronically infected rhesus macaques. They mapped the binding of these antibodies on SIVmac239 envelope trimer by binding competition assays with SIVmac239 Env-derived peptides, using loop-deleted Envs, and with EMPEM mapping. The antibodies mapped to V1/V2, V3, V4 and a glycan hole centered at residue 254. A cryo-EM structure of a SIVmac239 in complex with the antibody K11 was determined. K11 bound to the same glycan hole targeted by the ITS90.03 antibody and was closely located to the well-characterized glycan hole present on BG505SOSIP.664 trimer. Overall, the SIVmac239 had similar topology as SIVcpz and HIV-1 Env (BG505), although important differences were seen in the V1 and V4 loops, as well as the V5 loop, gp41 HR1, HR2, and fusion peptide. Experimental data was combined with computational analysis

to assess the extent of the glycan shield, and the poor or delayed neutralizing antibody responses in SIVmac239-infected macaques was attributed to more effective glycan shielding of SIVmac239 Env protein epitopes. This is a comprehensive study providing key knowledge on the SIVmac239 model, including well-characterized antibodies, a stabilized soluble trimeric Env, its cryo-EM structure, and characterization of the SIVmac239 glycan shield.

A few comments:

1. The V1 loop conformation of SIVmac239 is intriguing, although it is not clear how the coordinates for this loop, especially the stretch between residues 127 and 139, was built. The authors have stated that this region had poor resolution in the cryo-EM reconstruction, yet this region is included in the models. Cryo-EM reconstructions of this region should be shown, even if at lower contours, overlaid with the fitted model.
2. From looking at the structure figure in the left panel in Figure 4A, it appears that the SIVmac239 Env may have a less tightly packed trimer core than the BG505 (and other) HIV Env SOSIPs. Analysis of the interprotomer interactions, and comparing to known Env structures, should provide interesting insights.
3. A figure showing details of the 3D variability analysis (currently shown briefly in Supplementary Figure 4) and the scissoring motion that the authors refer to in the text, will be very interesting and useful to add.
4. The authors mention that K11 IgG was used for cryo-EM. Typically, Fabs are used for cryo-EM specimen preparation. Any particular reason IgG was used here instead of Fab? Might this have contributed to the considerable aggregation seen in the micrograph in Supplementary Figure 4a)?

Reviewer #3 (Remarks to the Author):

The manuscript by Zhao et al. describes the interactions of neutralizing antibodies (nAbs) with the envelope glycoproteins of simian immunodeficiency virus (SIV) and the ability of these nAbs to protect rhesus macaques from SIV infection. The authors identified 3 plasmas of rhesus macaques chronically infected with the pathogenic SIVmac239 that neutralized *in vitro* SIVmac239 pseudovirus infection. Twelve isolated nAbs from these sera could be clustered into 3 clonally related lineages, two of which contained long third complementarity determining region of the heavy chain Abs that is typical to many broadly neutralizing antibodies isolated from people living with HIV-1. The authors report the 3.4Å resolution cryo-EM structure of one nAb, K11 IgG, with the SIVmac239.K180S SOSIP trimer and identified the binding sites of antibodies from the 3 different lineages, which mapped to the V4 loop vicinity and to a glycan hole that is centered around K254 and atypical in many SIV strains. Analysis of the glycosylation pattern suggested that the SIVmac239.K180S SOSIP glycan shield may be denser than the one of SIVcpz or HIV-1 Env and may partially account for the poor nAb response of rhesus macaques to SIVmac239. Elegant challenge experiments in rhesus macaques showed that infusion of one antibody, K11, protected the animals from multiple intravenous SIVmac239 infections. The study is very comprehensive, important and well-designed; the research work provides high quality results and insights into the molecular interactions of nAbs and SIVmac239. Several points that could improve the manuscript follow.

Major points

1. If samples from the related rhesus macaques are still available, it will be very helpful to analyze contemporary SIV strains in rhesus macaques from which the 3 neutralizing plasmas were isolated. Are these viruses resistant to the isolated antibodies and are resistant mutations mapped to their binding sites? Similarly, did SIV viruses that infected the rhesus macaques despite K11 antibody infusion during the challenge experiment (Fig. 6) develop resistance to the K11 antibody?
2. Can the authors measure development of anti-K11 (anti-idiotypic) antibodies after infusion to the tested animals?
3. It is important to study the nAb response to SIV infection in rhesus macaques in order to better understand the development of an immune response. However, SIV infection of rhesus macaques

is not necessary the preferred model for testing synergy between nAb and cytotoxic T lymphocyte responses. There are several advanced SHIVs, including the pathogenic SHIV1054 that is based on transmitted/founder Env (Del Prete et al., *Cell Host & Microbe* 16, 412–418, 2014) and may provide more relevant context for HIV-1 transmission. These SHIVs robustly replicate in rhesus macaques and can likely be used in combination with many bnAbs and the CMV-based vectors developed by Louis Picker to test bnAb synergy with the cellular-mediated immune response. The authors should consider rephrasing the complete concept as it is currently stated, which is repeated in the introduction and discussion sections, and providing complete comparison between the potential future use of SIV vs SHIV as experimental models.

4. Correlates of protection mediated by SIV_DeltaNef vaccine have been identified and associated with non-neutralizing anti trimeric gp41 antibodies that were concentrated at mucosal front lines (Adnan et al., *PLoS Pathog* 2016; Voss et al., *AIDS* 2016). The authors should at least discuss the different alternatives of potential protection that are currently known. Also, what was the level of the infused K11 antibody in the mucosal tissues?

5. Can the authors compare the V1 loop length and glycan shield of SIVmac239 with those of the “easy-to-neutralize” SIVmac316?

Minor points

1. The study tested the protection from intravenous challenges of SIVmac239, maybe because the infection is more efficient than intravaginal or intrarectal infection but this route does not represent the most frequent way of HIV-1 transmission. Can the authors discuss the experiment setting and implications?

2. Sorting memory B cells that bind soluble SOSIP Env trimers will not detect B cells that express antibodies against the gp41 portion that is not expressed in the SOSIP and may miss other antibodies that do not bind well the soluble SOSIP. These limitations should be added to any general statement about the overall frequency of neutralizing/non-neutralizing antibodies in rhesus macaques that is solely based on binding to soluble SOSIP Env trimers (lines 127-128).

3. Line 98 – please define the AID50 abbreviation.

4. Supplementary Fig. 2 - in panel 2a the key has overlapping values (maybe because of rounding up the numbers) and the value “75” appears once with red background and twice with yellow background. Panel 2b - standard deviation was calculated from 2 (duplicate) measurements, which unreliably describe the deviation. Please consider reporting the range instead of standard deviation for only 2 measurements.

5. Please change “Infected sera” to “sera of SIVmac239-infected rhesus macaques” as well as “non-infected serum” to “serum of non-infected rhesus macaques” throughout the manuscript

6. Fig 4e – K11 antibody neutralizes SIVmac239 pseudoviruses 5-fold more efficiently than ITS90.03 antibody but seems to interact with smaller surface area according to the structural studies. can the authors discuss these differences?

7. Fig 6a – please add “Days” to scheme to clarify the time frame

8. Fig 6 - panel c is confusing because it shows different time frames for different animals. Please change the X-axis label to specify that it represents days post productive infection of each macaque.

9. Line 912 – it seems that a verb is missing in the sentence.

10. Line 941 – please revise “200,000 cells/well” as usually ~20,000 TZM-bl cells saturate one well of 96-well plate.

Reviewers' Comments:

Reviewer #1: The authors provide a very inclusive study of antibody mediated protection in the SIV model. Their studies of SIVmac239 infection of macaques is the best model for human HIV type-1 infection. Their approach using cryo-EM structures of the SIV envelope trimer appears highly likely to provide insights for vaccine design for the HIV-1 envelope, and surely further critical understanding of broadly neutralizing antibodies (bnAbs) in humans that have possible clinical application. Their major goal appears to address an important question directed to future HIV-1 virology. That is for their model, the use of nAb K11 as a passive reagent to be combined with T cell responses to investigate synergy of humoral and cellular protection against a SIVmac239 challenge.

The isolation and characterization of 12 potent SIVmac239 monoclonal nAbs from chronically infected macaques was well conceived and clearly presented. The identification of K11 neutralizing antibody and its subsequent use in the cryo-EM of the gp140 structure was probably the major accomplishment in this paper. The authors should again cross-check the data including peptide sequences in all figures and their legends as mentioned below. Is the identification of 7 mAbs to non-neutralizing SIVmac239 epitopes that are immunodominant related to the same situation in humans, potentially a discussion point for new readers not familiar with antibodies identified in early and late infected humans by HIV-1?

We thank the reviewer for the positive comments. We have cross-checked data and fixed where appropriate. Overall, the strong non-nAb responses to SIVmac239 infection parallel those seen in humans and presumably arise from strong antibody responses to non-functional forms of Env such as gp41 stumps (often referred to as viral debris). Some of the details are however different from HIV infection. The immunodominant non-nAb epitope specificities for SIVmac239 infection are the V4/V5 and V3 loops (based on mAb characterization) and gp41 (based on EMPEM analysis). The V3 loop and gp41 non-nAb specificities are similar to those observed for HIV. The V4 and V5, regions however, are neutralizing epitopes for some HIV strains but non-neutralizing for SIVmac239. This observation may be related to more glycan shielding on the SIVmac239 V4/V5 loop relative to many HIV strains. We have now made mention of this observation in lines 487-492 in the discussion.

The cryo-EM data of the SIVmac239 env trimer in complex with the nAb K11 was a major accomplishment, particularly at an overall 3.4 Å resolution. The presentation of the structural data was excellent. The extensive mapping of the glycans onto the trimer, a major effort by itself, was also well presented. The authors present a compelling argument for glycan content of the trimer and its relationship to low immunogenicity.

The studies which show protection of rhesus macaques by nAb from repeated SIVmac239 challenge highly suggest humoral immunity plays an important role in protection. The literature cited was excellent and inclusive.

Minor points:

The manuscript is littered with a lot of typos maybe because of contributions from many investigators from various areas. Some are listed below. The authors should be sure that the sequences of the peptides listed in all Figures and Tables are correct. Correlation of what is the text and that information that is in the figures and legends should also be checked again.

Line 179, space between (Fig. 2c,dand...) space between d/and.

Line 189, Fig. 2d should read 2e

Line 1021, Page 35: "guassian blot template" should be "gaussian plot template"

We thank the reviewer for identifying these typos, which we have now fixed. We have also performed a more thorough copy/edit to correct these mistakes. With regard to the last comment, the word should be “blob”.

Reviewer #2: In this paper titled “Molecular insights into antibody-mediated protection against the prototypic simian immunodeficiency virus”, Zhao et al. have isolated 12 potent nAbs against SIVmac239 virus from three chronically infected rhesus macaques. They mapped the binding of these antibodies on SIVmac239 envelope trimer by binding competition assays with SIVmac239 Env-derived peptides, using loop-deleted Envs, and with EMPEM mapping. The antibodies mapped to V1/V2, V3, V4 and a glycan hole centered at residue 254. A cryo-EM structure of a SIVmac239 in complex with the antibody K11 was determined. K11 bound to the same glycan hole targeted by the ITS90.03 antibody and was closely located to the well-characterized glycan hole present on BG505SOSIP.664 trimer. Overall, the SIVmac239 had similar topology as SIVcpz and HIV-1 Env (BG505), although important differences were seen in the V1 and V4 loops, as well as the V5 loop, gp41 HR1, HR2, and fusion peptide. Experimental data was combined with computational analysis to assess the extent of the glycan shield, and the poor or delayed neutralizing antibody responses in SIVmac239-infected macaques was attributed to more effective glycan shielding of SIVmac239 Env protein epitopes. This is a comprehensive study providing key knowledge on the SIVmac239 model, including well-characterized antibodies, a stabilized soluble trimeric Env, its cryo-EM structure, and characterization of the SIVmac239 glycan shield.

A few comments:

1. The V1 loop conformation of SIVmac239 is intriguing, although it is not clear how the coordinates for this loop, especially the stretch between residues 127 and 139, was built. The authors have stated that this region had poor resolution in the cryo-EM reconstruction, yet this region is included in the models. Cryo-EM reconstructions of this region should be shown, even if at lower contours, overlaid with the fitted model.

The quality in this region is indeed poorer than the rest of the map. However, at lower resolutions and contour levels, the map density becomes fully resolved and connected. Although model building would not typically be performed in such regions, given the uniqueness of this loop, we felt it would be particularly useful to include in the model. The loop itself was modeled de-novo then further refined automatically with the ROSETTA fragment-based refinement and relax functions. To further clarify the relative modeling confidence in this region, a per-residue RMSD calculated from 100 unique models generated by ROSETTA has now been added to Supplemental Figure 4 along with images showing the map density in this region at high and low contour along with a gaussian filtered map.

2. From looking at the structure figure in the left panel in Figure 4A, it appears that the SIVmac239 Env may have a less tightly packed trimer core than the BG505 (and other) HIV Env SOSIPs. Analysis of the interprotomer interactions, and comparing to known Env structures, should provide interesting insights.

A comprehensive molecular interface analysis was performed for SIVmac239, SIVcpz (MT145K), and HIV-1 Env (BG505 SOSIP.v5.2) with PDBePISA. The results are displayed in Figure 1 accompanying this review (see below). Both the SIVmac and BG505 structures have a complete gp41 modeled, however, the SIVcpz structure is missing HR1, which affects the gp41-gp41 interface analysis. Despite this, the analysis shows that the interfaces are essentially

equivalent. The gp120-gp41 interface of SIVmac is actually the largest, while its gp120-gp120 interface is slightly smaller than HIV and SIVcpz, but only by $\sim 100 \text{ \AA}^2$. We do not think this analysis provides anything particularly useful to the manuscript and have chosen not to include it in the revised draft. We hope the reviewer finds the analysis satisfactory.

3. A figure showing details of the 3D variability analysis (currently shown briefly in Supplementary Figure 4) and the scissoring motion that the authors refer to in the text, will be very interesting and useful to add.

A supplemental movie has now been added to the manuscript showing the two primary modes of variability.

4. The authors mention that K11 IgG was used for cryo-EM. Typically, Fabs are used for cryo-EM specimen preparation. Any particular reason IgG was used here instead of Fab? Might this have contributed to the considerable aggregation seen in the micrograph in Supplementary Figure 4a)?

It is true that Fabs are typically used, however, in this case, we were having trouble achieving high-enough binding stoichiometry to obtain a structure, which we were able to overcome by using the complete IgG. This did result in the dimerization of trimers by the two Fab arms binding different molecules, however, this may have actually facilitated the reconstruction process by helping to overcome the extreme orientation bias that previously prevented us from obtaining a high-resolution trimer structure, presumably by blocking the charged base of the trimer from the air-water interface. This is the "aggregation" seen in the micrograph.

Reviewer #3: The manuscript by Zhao et al. describes the interactions of neutralizing antibodies (nAbs) with the envelope glycoproteins of simian immunodeficiency virus (SIV) and the ability of these nAbs to protect rhesus macaques from SIV infection. The authors identified 3 plasmas of rhesus macaques chronically infected with the pathogenic SIVmac239 that neutralized in vitro SIVmac239 pseudovirus infection. Twelve isolated nAbs from these sera could be clustered into 3 clonally related lineages, two of which contained long third complementarity determining region of the heavy chain Abs that is typical to many broadly neutralizing antibodies isolated from people living with HIV-1. The authors report the 3.4A resolution cryo-EM structure of one nAb, K11 IgG, with the SIVmac239.K180S SOSIP trimer and identified the binding sites of antibodies from the 3 different lineages, which mapped to the V4 loop vicinity and to a glycan hole that is centered around K254 and atypical in many SIV strains.

Analysis of the glycosylation pattern suggested that the SIVmac239.K180S SOSIP glycan shield may be denser than the one of SIVcpz or HIV-1 Env and may partially account for the poor nAb response of rhesus macaques to SIVmac239. Elegant challenge experiments in rhesus macaques showed that infusion of one antibody, K11, protected the animals from multiple intravenous SIVmac239 infections. The study is very comprehensive, important and well-designed; the research work provides high quality results and insights into the molecular interactions of nAbs and SIVmac239. Several points that could improve the manuscript follow.

Major points:

1. If samples from the related rhesus macaques are still available, it will be very helpful to analyze contemporary SIV strains in rhesus macaques from which the 3 neutralizing plasmas were isolated. Are these viruses resistant to the isolated antibodies and are resistant mutations mapped to their binding sites? Similarly, did SIV viruses that infected the rhesus macaques despite K11 antibody infusion during the challenge experiment (Fig. 6) develop resistance to the

K11 antibody?

Unfortunately, the samples from these animals are very limited, and we are therefore unable to perform virus sequencing for these animals. As shown in Figure 6 and Supplementary Figure 7, following a single Ab infusion and three weekly sequential challenges, the first virus breakthrough occurred at day 21 / wk 3 when the Ab titers are at their lowest serum concentrations. A second infusion at day 21 / wk3 protected the remaining animals for four weeks despite continued weekly challenges. These data suggest that breakthrough occurs when nAb titers are low and are not due to selection of resistant virus variants.

2. Can the authors measure development of anti-K11 (anti-idiotypic) antibodies after infusion to the tested animals?

We do not have an anti-K11 idiotype antibody to reliably measure ADA responses following antibody infusion. Instead, we measured rhesus plasma binding to biotinylated K11 Fab by ELISA to assess the level of anti-K11 responses. As shown in Figure 2 accompanying this review below, many of the macaques developed ADA response by day 49 after two antibody infusions. We do not see a strong trend of ADA responses and time to infection. For example, animals r17041 (became infected after 3rd challenge between day 14 and day 21) and r17034 (became infected after 8th challenge between day 49 and 56) both developed comparable ADA responses but had different outcomes in terms of time to infection post-antibody infusion. We therefore conclude that neutralizing antibody concentration is still the key determinant for viral infection.

3. It is important to study the nAb response to SIV infection in rhesus macaques in order to better understand the development of an immune response. However, SIV infection of rhesus macaques is not necessary the preferred model for testing synergy between nAb and cytotoxic T lymphocyte responses. There are several advanced SHIVs, including the pathogenic SHIV1054 that is based on transmitted/founder Env (Del Prete et al., Cell Host & Microbe 16, 412–418, 2014) and may provide more relevant context for HIV-1 transmission. These SHIVs robustly replicate in rhesus macaques and can likely be used in combination with many bnAbs and the CMV-based vectors developed by Louis Picker to test bnAb synergy with the cellular-mediated immune response. The authors should consider rephrasing the complete concept as it is currently stated, which is repeated in the introduction and discussion sections, and providing complete comparison between the potential future use of SIV vs SHIV as experimental models.

We agree that the SIV and SHIV models are complementary. While SHIVs enable study of HIV Env in antibody-mediated protection studies, there are also some known caveats. SHIVs do not readily replicate in rhesus macaques without engineering the CD4bs receptor for higher affinity for rhesus CD4 and/or through serial passage in primates. The STEP trial also showed no protection efficacy in clinical trials, which is consistent with SIV studies and not reflected in corresponding studies with SHIV. SIV is considered more reflective of the genotype diversity of HIV than SHIV and is therefore more suited for evaluation of T cell-based vaccines. To this point, the rhesus CMV developed by Louis Picker was designed to elicit SIV-specific T cell responses and showed control of SIVmac239-infected animals. We are not arguing against the value of SHIVs but maintain the point that these SIVmac239 nAbs enable ready evaluation of synergistic contributions of antibody and cellular responses to prevent SIV infection through combination with Louis Picker's rhCMV platform, which has a wealth of data for the highly pathogenic SIVmac239 isolate. Similar studies with SHIV isolates, although possible, would

require significant investments and studies to generate similar data packages. We have included these points in lines 431-450 in the discussion.

4. Correlates of protection mediated by SIV_DeltaNef vaccine have been identified and associated with non-neutralizing anti trimeric gp41 antibodies that were concentrated at mucosal front lines (Adnan et al., PLoS Pathog 2016; Voss et al., AIDS 2016). The authors should at least discuss the different alternatives of potential protection that are currently known. Also, what was the level of the infused K11 antibody in the mucosal tissues?

We have shown in this manuscript that passive transfer of a high dose of non-neutralizing antibodies did not result in protection against virus challenge. The Voss et al manuscript on anti-trimeric gp41 antibodies did not include virus challenge and is therefore a hypothesis of a correlate of protection and not a confirmed correlate. More to this point, non-neutralizing anti-gp41 antibody elicited at mucosal front lines is a possible correlate of SIVdelNef vaccine-mediated protection, but was not confirmed to be a mechanistic correlate of protection (e.g. via passive transfer studies and subsequent challenge) and does not exclude the contribution of nAbs to protection because none were elicited (Li et al, JI 2014). We have included lines to mention these other studies (lines 500-504) but because these other data are largely hypothetical, we elect to focus the introduction and discussion on the data generated from these studies and the conclusions derived thereof. We did not collect mucosal secretions from this study and are not able to measure antibody concentrations in the mucosa.

5. Can the authors compare the V1 loop length and glycan shield of SIVmac239 with those of the “easy-to-neutralize” SIVmac316?

The V1 loop length and glycan shield (based on the total number of glycan sequons) are not different between SIVmac239 and SIVmac316. Based on previous studies (Means et al, JVI, 2001), the small number of mutations in SIVmac316 renders the virus sensitive to antibodies to the V3 loop. Based on these data, we hypothesize that these mutations destabilize the V1/V2 loop and lead to an open Env conformation and therefore greater exposure of the V3 loop, which subsequently leads to a more neutralization-sensitive phenotype. The loop length of the two clones is identical and de-novo modeling by AlphaFold2 did not reveal any difference between the structures in this region or elsewhere (data not shown).

Minor points

1. The study tested the protection from intravenous challenges of SIVmac239, maybe because the infection is more efficient than intravaginal or intrarectal infection but this route does not represent the most frequent way of HIV-1 transmission. Can the authors discuss the experiment setting and implications?

While IV challenge is not the most frequent way of HIV-1 transmission, it is a very stringent challenge compared to intravaginal and intrarectal, which further highlights the contributions of nAbs to protection. We have included this line in the discussion (lines 454-456). The experiment setting is same as Fuchs et al, Plos Path 2015 where 5L7 was included in the challenge study. The same PBMC-grown stock of cloned SIVmac239 was used in our study, which has been carefully titered previously by the IV route in monkeys and has been used extensively by numerous studies (Lewis et al, 1994; Lifson et al, JVI 2001; Jia et al, Plos Path 2009; etc).

2. Sorting memory B cells that bind soluble SOSIP Env trimers will not detect B cells that express antibodies against the gp41 portion that is not expressed in the SOSIP and may miss other antibodies that do not bind well the soluble SOSIP. These limitations should be added to

any general statement about the overall frequency of neutralizing/non-neutralizing antibodies in rhesus macaques that is solely based on binding to soluble SOSIP Env trimers (lines 127-128).

The reviewer is correct with regard to the absence of a portion of gp41 in the SOSIP construct. Our comparison of nAb/non-nAb frequency between SIVmac239 infected and SHIV_{BG505} infected macaques is calculated based on SOSIP sorting (SIVmac239 SOSIP vs BG505 SOSIP) – from this comparison, we also find that SIVmac239-infected animals have lower nAb frequency than BG505-SHIV infected animals. The comparison was added in line 127 in the text.

3. Line 98 – please define the AID50 abbreviation.

The definition for AID50 is now added to line 98.

4. Supplementary Fig. 2 - in panel 2a the key has overlapping values (maybe because of rounding up the numbers) and the value “75” appears once with red background and twice with yellow background. Panel 2b - standard deviation was calculated from 2 (duplicate) measurements, which unreliably describe the deviation. Please consider reporting the range instead of standard deviation for only 2 measurements.

The color now has been clarified in Supplementary Fig. 2a so that all “75” values are with yellow background.

5. Please change “Infected sera” to “sera of SIVmac239-infected rhesus macaques” as well as “non-infected serum” to “serum of non-infected rhesus macaques” throughout the manuscript

The “infected sera” now has been changed to “SIVmac239-infected macaque sera” throughout the manuscript.

6. Fig 4e – K11 antibody neutralizes SIVmac239 pseudoviruses 5-fold more efficiently than ITS90.03 antibody but seems to interact with smaller surface area according to the structural studies. can the authors discuss these differences?

In general, it is difficult to correlate epitope properties with neutralization efficiency, however, one could speculate that the purely gp120 epitope of K11, as opposed to the combined gp41/gp120 epitope of ITS90.03, may have some effect along these lines, as gp41 and the gp41/gp120 interface are more flexible than the gp120 core. We have included this discussion in lines 298-301.

7. Fig 6a – please add “Days” to scheme to clarify the time frame

“Days” is added to Fig. 6a scheme.

8. Fig 6 - panel c is confusing because it shows different time frames for different animals. Please change the X-axis label to specify that it represents days post productive infection of each macaque.

The X axis label in Fig. 6c is now changed to “Days post infection”.

9. Line 912 – it seems that a verb is missing in the sentence.

We have modified this sentence for clarity.

10. Line 941 – please revise “200,000 cells/well” as usually ~20,000 TZM-bl cells saturate one well of 96-well plate.

We thank the reviewer for identifying this typo and have revised as “10,000 cells/well”.

A

■ SIVmac239
■ SIVcpz MT145K
■ HIV-1 BG505

B

SIVmac239 SOSIP			BG505 SOSIPv5.2 (PDBID: 7L8T)			SIVcpz MT145K SOSIP (PDBID: 6OHY)		
Interface Summary		XML	Interface Summary		XML	Interface Summary		XML
Selection range	Structure 1	Structure 2	Selection range	Structure 1	Structure 2	Selection range	Structure 1	Structure 2
class	A	B	class	E	F	class	C	E
symmetry operation	x,y,z	--	symmetry operation	x,y,z	x,y,z	symmetry operation	x,y,z	x,y,z
symmetry ID	1_555	0_555	symmetry ID	1_555	0_555	symmetry ID	1_555	0_555
Number of atoms			Number of atoms			Number of atoms		
interface	234 8.9%	288 26.2%	interface	246 7.0%	285 23.1%	interface	260 7.0%	262 27.2%
surface	2258 87.1%	851 74.8%	surface	2177 61.9%	865 76.4%	surface	2252 63.5%	785 79.4%
total	3953 100.0%	1138 100.0%	total	3516 100.0%	1147 100.0%	total	3547 100.0%	964 100.0%
Number of residues			Number of residues			Number of residues		
interface	62 12.6%	79 55.6%	interface	68 15.3%	72 60.0%	interface	65 14.0%	70 68.3%
surface	487 94.7%	141 99.3%	surface	434 97.6%	144 100.0%	surface	442 98.9%	120 100.0%
total	493 100.0%	142 100.0%	total	445 100.0%	144 100.0%	total	447 100.0%	120 100.0%
Solvent-accessible area, Å ²			Solvent-accessible area, Å ²			Solvent-accessible area, Å ²		
interface	2895.1 11.0%	2541.9 23.8%	interface	2682.0 11.7%	2578.7 22.4%	interface	2835.1 10.9%	2700.0 25.7%
total	24493.3 100.0%	11081.7 100.0%	total	22831.0 100.0%	11516.4 100.0%	total	24073.8 100.0%	10515.8 100.0%
Solvation energy, kcal/mol			Solvation energy, kcal/mol			Solvation energy, kcal/mol		
isolated structure	-454.1 100.0%	-65.1 100.0%	isolated structure	-423.2 100.0%	-93.8 100.0%	isolated structure	-414.4 100.0%	-77.3 100.0%
gain on complex formation	-24.9 5.5%	-16.5 17.3%	gain on complex formation	-23.5 5.6%	-21.0 22.4%	gain on complex formation	-19.3 4.7%	-19.5 25.3%
average gain	-3.3 0.7%	-15.4 17.2%	average gain	-1.9 0.4%	-14.5 15.5%	average gain	-3.2 0.8%	-16.0 20.7%
P-value	0.000	0.501	P-value	0.000	0.118	P-value	0.000	0.259

gp120-gp41 interface			gp120-gp120 interface			gp41-gp41 interface		
Interface Summary		XML	Interface Summary		XML	Interface Summary		XML
Selection range	Structure 1	Structure 2	Selection range	Structure 1	Structure 2	Selection range	Structure 1	Structure 2
class	A	C	class	C	E	class	A	C
symmetry operation	x,y,z	--	symmetry operation	x,y,z	x,y,z	symmetry operation	x,y,z	x,y,z
symmetry ID	1_555	0_555	symmetry ID	1_555	0_555	symmetry ID	1_555	0_555
Number of atoms			Number of atoms			Number of atoms		
interface	39 1.0%	34 0.9%	interface	39 1.1%	59 1.7%	interface	39 1.1%	59 1.4%
surface	2258 57.1%	2263 57.0%	surface	2208 62.8%	2177 61.6%	surface	2252 63.5%	2252 63.8%
total	3953 100.0%	3953 100.0%	total	3516 100.0%	3516 100.0%	total	3547 100.0%	3547 100.0%
Number of residues			Number of residues			Number of residues		
interface	14 2.8%	9 1.8%	interface	8 1.8%	15 3.4%	interface	9 2.0%	15 3.4%
surface	487 94.7%	467 94.7%	surface	439 98.7%	434 97.5%	surface	441 98.7%	442 98.9%
total	493 100.0%	493 100.0%	total	445 100.0%	445 100.0%	total	447 100.0%	447 100.0%
Solvent-accessible area, Å ²			Solvent-accessible area, Å ²			Solvent-accessible area, Å ²		
interface	334.8 1.4%	354.7 1.4%	interface	458.1 2.0%	413.5 1.8%	interface	420.2 1.7%	597.8 1.7%
total	24493.3 100.0%	24493.9 100.0%	total	23050.6 100.0%	22831.0 100.0%	total	24067.7 100.0%	24073.5 100.0%
Solvation energy, kcal/mol			Solvation energy, kcal/mol			Solvation energy, kcal/mol		
isolated structure	-454.1 100.0%	-454.4 100.0%	isolated structure	-419.0 100.0%	-423.2 100.0%	isolated structure	-414.4 100.0%	-414.4 100.0%
gain on complex formation	-1.9 0.4%	-1.2 0.3%	gain on complex formation	-2.0 0.5%	-1.7 0.4%	gain on complex formation	-2.2 0.5%	-1.7 0.4%
average gain	-0.5 0.1%	-0.5 0.1%	average gain	-0.4 0.1%	-0.5 0.1%	average gain	-0.5 0.1%	-0.8 0.2%
P-value	0.245	0.364	P-value	0.196	0.303	P-value	0.196	0.336

gp120-gp41 interface			gp41-gp41 interface		
Interface Summary		XML	Interface Summary		XML
Selection range	Structure 1	Structure 2	Selection range	Structure 1	Structure 2
class	E	F	class	B	D
symmetry operation	x,y,z	--	symmetry operation	x,y,z	x,y,z
symmetry ID	1_555	0_555	symmetry ID	1_555	0_555
Number of atoms			Number of atoms		
interface	142 12.5%	137 12.0%	interface	110 9.2%	108 9.4%
surface	859 75.5%	853 75.0%	surface	905 75.5%	849 74.0%
total	1138 100.0%	1138 100.0%	total	1198 100.0%	1147 100.0%
Number of residues			Number of residues		
interface	30 21.1%	38 28.8%	interface	34 22.4%	27 18.8%
surface	141 99.3%	141 99.3%	surface	152 100.0%	144 100.0%
total	142 100.0%	142 100.0%	total	152 100.0%	144 100.0%
Solvent-accessible area, Å ²			Solvent-accessible area, Å ²		
interface	1387.8 12.6%	1337.1 12.1%	interface	1156.8 9.4%	1151.7 10.0%
total	11090.7 100.0%	11089.8 100.0%	total	12245.3 100.0%	11463.6 100.0%
Solvation energy, kcal/mol			Solvation energy, kcal/mol		
isolated structure	-95.0 100.0%	-95.2 100.0%	isolated structure	-101.7 100.0%	-95.4 100.0%
gain on complex formation	-6.0 8.4%	-11.3 11.9%	gain on complex formation	-11.9 11.7%	-6.2 6.5%
average gain	-7.7 8.1%	-7.5 7.9%	average gain	-6.7 8.6%	-5.9 6.1%
P-value	0.449	0.157	P-value	0.057	0.441

gp120-gp41 interface

gp120-gp120 interface

gp41-gp41 interface

Review Figure 1 | Molecular interface analysis. (A) Atomic models of SIVmac239, SIVcpz, and HIV-1 Env superimposed. (B) PDBePISA analysis for single gp120-gp41, gp120-gp120, and gp41-gp41 interfaces.

Review Figure 2 | Plasma Anti-K11 Antibody ELISA. Rhesus plasma samples from the K11 group were measured by ELISA binding to biotinylated K11 Fab. Alkaline phosphatase-conjugated anti-human IgG Fc secondary antibody was then added and absorbance was measured at OD405 nm. Animal r18011 sample from the 5L7 group served as negative control. In the positive control, alkaline phosphatase-conjugated anti-human IgG (Fab')₂ antibody was used as secondary antibody.

Reviewer comments, second round -

Reviewer #2 (Remarks to the Author):

The authors have satisfactorily addressed all my critiques.

Reviewer #3 (Remarks to the Author):

The authors satisfactorily addressed most of my comments in the revised manuscript and the study described will be valuable and very helpful to the HIV-1 vaccine field.